# 🍸 Tequila: Trapping-free Ternary Quantization for Large Language Models

**Hong Huang**[1][*] **Decheng Wu**[2] **Rui Cen**[2] **Guanghua Yu**[2] **Zonghang Li**[3]
**Kai Liu**[2] **Jianchen Zhu**[2] **Peng Chen**[2] **Xue Liu**[4] **Dapeng Wu**[1]
[1]City University of Hong Kong    [2]Tencent    [3]MBZUAI    [4]McGill University

## Abstract

Quantization techniques are essential for the deployment of Large Language Models (LLMs) on edge devices. However, prevailing methods often rely on mixed-precision multiplication that lacks efficient hardware support, making it not feasible. Ternary weight quantization addresses this by constraining weights to {-1, 0, 1}, replacing expensive multiplications with hardware-efficient additions. However, such aggressive compression leads to significant accuracy degradation, even after costly quantization-aware training with massive data. We identify the core issue as ***deadzone trapping:*** *a large number of weights are trapped at the deadzone boundary.* This occurs because these weights receive only noisy, less informative gradients, preventing stable escape from the deadzone and severely impeding model capacity and optimization. To address this issue, we propose **Tequila**, a trapping-free quantization optimization method that reactivates deadzone-trapped weights by repurposing them as dynamic biases. This allows the repurposed weights to provide a continuous signal in the forward pass and, critically, receive direct, meaningful gradient signals during backpropagation, thereby enhancing model capacity and optimization with nearly *zero* inference overhead. Extensive evaluations demonstrate that Tequila outperforms state-of-the-art (SOTA) ternary quantization methods across five benchmarks. Specifically, on the ARC benchmark, it achieves $> 4\%$ accuracy gain over the SOTA baseline, nearly matching full-precision performance (within $< 1\%$ gap) with a $3.0\times$ inference speedup. Consequently, Tequila offers a practical and efficient implementation for the deployment of advanced LLMs in resource-constrained environments. The code is available at `https://github.com/Tencent/AngelSlim/tree/tequila/TernaryQuant`.

## 1 Introduction

Recent advancements in large language models (LLMs) (Wu et al., 2023; Floridi & Chiriatti, 2020; Zhang et al., 2022) have demonstrated remarkable success across a wide range of applications, from conversational chatbots to creative writing. However, growing concerns over data privacy, the need for offline functionality, and the high cost of large-scale cloud deployment (Yao et al., 2024; Liagkou et al., 2024) have necessitated the deployment of these models directly on edge devices, which are usually resource-constrained.

Quantization (Lin et al., 2023; Frantar et al., 2022) has emerged as a promising technique to achieve this goal, reducing model size and computational requirements by representing model weights with lower precision. However, most existing quantization methods (Kwon et al., 2022; Lin et al., 2023; Frantar et al., 2022) are primarily designed for server-grade GPUs that support specialized hardware features, such as mixed-precision multiplication (Lin et al., 2023). These methods are often incompatible with edge and mobile hardware, highlighting a critical need for hardware-friendly quantization approaches that remain effective across diverse and resource-constrained devices.

Ternary quantization (Li et al., 2016; Liu & Liu, 2023; Ma et al., 2025; Wang et al., 2023; 2025a) offers a promising path for on-device deployment of LLMs by constraining weights to $\{-1, 0, +1\}$.

---

[*]Work with Tencent

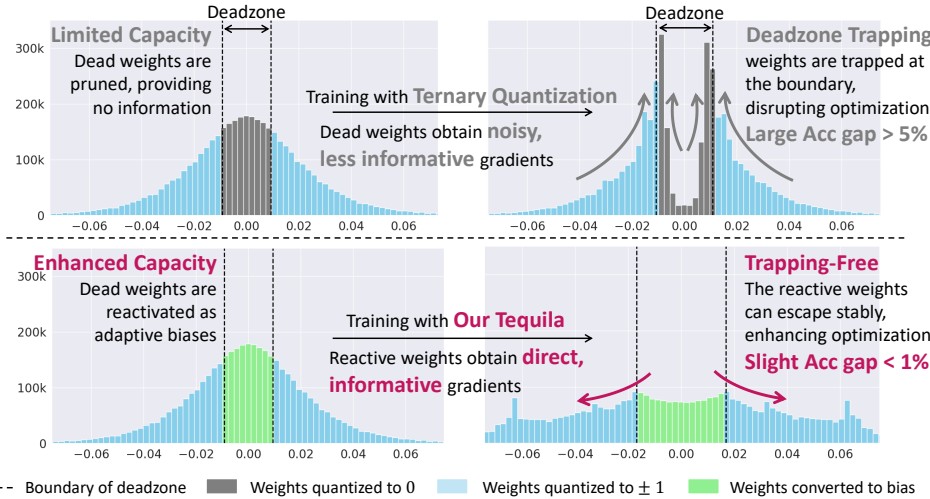

Figure 1: *(Top)* **Deadzone Trapping in Ternary Quantization**: Dead weights are trapped around the deadzone boundary due to noisy and less informative gradients, impeding model capacity and optimization, causing a significant accuracy drop ($> 5\%$) versus the full-precision. *(Bottom)* **Reactivation Strategy of Tequila**: Our Tequila reactivates dead weights as biases, providing meaningful gradients for stable escapes, enhancing model optimization, achieving a minor accuracy gap ($< 1\%$).

This method reduces matrix multiplication to efficient additions, as illustrated in Fig. 2, which are widely supported by most hardware. However, such aggressive compression introduces significant information loss, often leading to severe accuracy degradation even after costly Quantization-Aware Training (QAT) on massive datasets. For instance, BitNet (Ma et al., 2025; Wang et al., 2025a) consumes 4T tokens during QAT but still fails to match full-precision performance. Thus, the dual problems of performance degradation and prohibitive training overheads persist as the fundamental barriers to the development of effective ternary LLMs.

In this paper, we identify the key source of these challenges as **deadzone trapping**, where *a large number of weights are trapped at the deadzone boundary*. Deadzone trapping arises from the aggressive nature of ternary quantization, which creates a vast deadzone that quantizes a large proportion of weights to zero. During training, these "dead" weights receive only noisy, less informative gradients from the Straight-Through Estimator (STE), preventing effective optimization. Lacking consistent directional signals, these weights are hard to escape the deadzone stably and are accumulated at the deadzone boundary, as shown in Fig. 1 *(Top)*. This results in a cycle of ineffective oscillation, rendering these weights long-term inactive and severely impeding model capacity and optimization.

To address the deadzone trapping issue, we propose **Tequila**, a trapping-free **Te**rnary **qu**antization method for **la**rge language models. Our key idea is to reactivate dead weights by repurposing them as biases. This provides continuous signals to the output, enhancing the model capacity, as shown in Fig. 2 (c). More importantly, these weights receive direct and informative gradients via the bias terms, enabling them to escape the deadzone stably, as shown in Fig. 1 *(Bottom)*. Crucially, these biases can be computed offline, introducing nearly zero inference overhead.

We evaluate Tequila on five common benchmarks using LLaMA 3.2 (Touvron et al., 2023) and Qwen3 (Bai et al., 2023) models. Our experiments demonstrate that Tequila outperforms all state-of-the-art (SOTA) ternary methods across all benchmarks while requiring only limited training data. For instance, when trained on just 10B tokens, Tequila achieves a $> 4\%$ accuracy gain over the SOTA baseline on the ARC benchmark, nearly matching full-precision performance. Furthermore, it delivers a significant $3\times$ inference speedup on an Intel 8263C CPU, verifying that Tequila offers a practical and efficient solution for deploying LLMs on resource-constrained devices.

## 2 BACKGROUND AND CHALLENGE

### 2.1 TERNARY QUANTIZATION

Ternary quantization is an extreme compression technique that constrains model weights to ternary values $\{-1, 0, +1\}$. This representation converts the computationally expensive weight-input matrix

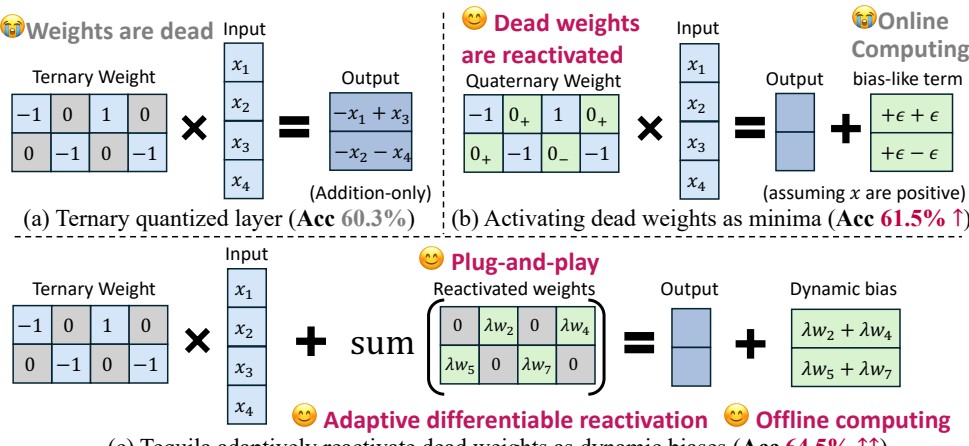

Figure 2: *(a)* Prior Ternary Quantization replaces multiplications with efficient additions but suffers from severe information loss and limited capacity due to deadzone-trapped weights. *(b)* Minima Reactivation assigns signed minima to dead weights, improving capacity but yielding only marginal accuracy gains. *(c)* Tequila reactivates dead weights as adaptive dynamic biases via a differentiable function, achieving significant accuracy improvements with nearly zero inference overhead. For simplicity, we omit the scaling operation in the Figure.

multiplication into input-inner addition, as shown in Fig. 2 (a), offering significant hardware advantages. Given a full-precision weight vector $W = (w_1, \ldots, w_n)$, the general form of the ternary quantization function $Q(\cdot)$ is defined as:

$$Q(W) = \hat{W}\alpha, \quad \hat{w}_i = \begin{cases} +1, & \text{if } w_i \geq \Delta; \\ 0, & \text{if } |w_i| < \Delta; \\ -1, & \text{if } w_i \leq -\Delta, \end{cases} \quad (1)$$

where $\hat{W} = (\hat{w}_1, \ldots, \hat{w}_n)$ is ternary weights, $\alpha$ is a scaling factor and $\Delta$ is a threshold parameter. A significant body of research focuses on determining optimal values for $\alpha$ and $\Delta$. For instance, the TWN (Li et al., 2016) assumes the weight distribution follows a standard Gaussian distribution. It approximates the optimal threshold as $\Delta^* \approx \frac{0.75}{n} \sum_{i=1}^{n} |w_i|$ and derives a closed-form solution for $\alpha$ by minimizing $||W - \alpha\hat{W}||^2$. Subsequent methods (Chen et al., 2024; Liu et al., 2025; Zhu et al., 2016) forgo this distributional hypothesis and instead treat $\alpha$ or $\Delta$ as trainable parameters learned during optimization. In recent open-source ternary LLMs (Ma et al., 2025; Team et al., 2025; Kaushal et al., 2025), the static absmean quantization method has gained wider adoption due to its training stability, where the $\alpha$ and $\Delta$ are defined by

$$\alpha = \frac{1}{n} \sum_{i=1}^{n} |w_i|, \quad \Delta = \frac{\alpha}{2}. \quad (2)$$

Due to the aggressive nature of this compression, quantized models often require Quantization-Aware Training (QAT) to recover accuracy. During QAT, full-precision weights are dynamically quantized using a quantization function $Q(\cdot)$ in Eq. 1 for the forward pass, while the backward pass operates on full-precision gradients. This process maintains a full-precision copy of the weights $W$ to accumulate gradient updates, as detailed in Appendix C. Due to a non-differentiable function of $Q(\cdot)$, the gradients for $W$ are approximated using the Straight-Through Estimator (STE) (Zhu et al., 2016; Chen et al., 2024), leading to the following forward pass and backpropagation with input vector $X = (x_1, \ldots, x_n)$:

$$Y = X^T Q(W) = X^T \hat{W}\alpha, \quad \frac{\partial L}{\partial w_i} = \begin{cases} \frac{\partial L}{\partial Y} x_i \alpha, & \text{if } |w_i| \geq \Delta; \\ \frac{\partial L}{\partial Y} x_i, & \text{if } |w_i| < \Delta, \end{cases} \quad (3)$$

where $L$ denotes the loss of the model prediction. After training, the full-precision weights $W$ are discarded. The ternary weights $\hat{W}$ and the scaling factor $\alpha$ are packed for inference. During inference, the ternary multiplication of $X^T \hat{W}$ is computed first, following the efficient process shown in Fig. 2 (a). This eliminates the need for mixed-precision matrix multiplication, replacing it with hardware-efficient addition. The related work and details can be found in Appendix B and E.4.

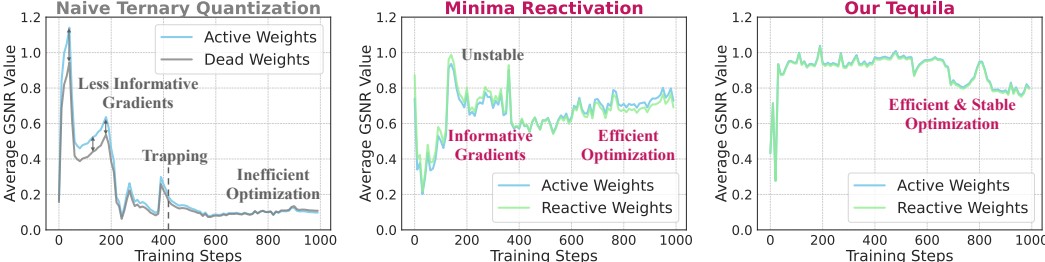

Figure 3: The average Gradient Signal-to-Noise Ratio (GSNR) for *(left)* Naive Ternary Quantization, *(middle)* Minima Reactivation, and *(right)* our Tequila method during training.

## 2.2 DEADZONE TRAPPING

Ternary quantization incurs significant information loss and performance degradation, necessitating extensive retraining to recover model accuracy. This problem is exacerbated in LLMs, where the scale of parameters amplifies the difficulty of retraining. For instance, BitNet (Ma et al., 2025) trains from scratch on 4T tokens, a cost rivaling standard pre-training. Similarly, BitCPM4 (Team et al., 2025) uses 100B tokens of training even when starting from a pre-trained model. To our knowledge, no existing work has achieved competitive performance with less than 100B tokens for ternary-quantized LLMs.

We identify the core cause of this inefficiency as **deadzone trapping**, *where a large number of weights become trapped at the deadzone boundary*. This issue originates from the aggressive nature of ternary quantization, which creates a deadzone within the range $(-\Delta, \Delta)$ where weights are quantized to zero. During Training, these "dead" weights ($\hat{w}_i = 0$) and their corresponding inputs $x_i$ are continually pruned in the forward pass, contributing nearly no information to the output $Y$ and the loss $L$. Consequently, the upstream signal $\frac{\partial L}{\partial Y}$ reflects less sensitivity to $w_i$ and $x_i$, making the overall gradients in Eq. 3 noisy and less informative. This problem is exacerbated by the non-differentiable quantization function $Q(\cdot)$, as the required use of the STE injects significant noise into these gradients. As shown in Fig. 3 *(left)*, the Gradient Signal-to-Noise Ratio (GSNR) for naive ternary quantization confirms this analysis: in early training, the GSNR for dead weights is much lower than for active weights (see Appendix E.2 for more details and analysis).

These noisy gradients prevent dead weights from accumulating consistent update signals, trapping them within the deadzone. When a large number of weights remain at deadzone for an extended period, the model easily becomes trapped in a suboptimal state. When some weights occasionally break free, their abrupt changes often trigger opposing gradients that pull them back. This dynamic results in weights accumulating at the deadzone boundary, trapped in a cycle of ineffective oscillation, as shown in Fig. 1 *(top)* and Fig. 10. Ultimately, deadzone trapping renders a substantial portion of the model weights long-term inactive, severely impairing both model capacity and training efficiency, as evidenced in Fig. 3 *(left)*.

## 3 TEQUILA: DEADZONE-FREE TERNARY QUANTIZATION

### 3.1 MITIGATING THE DEADZONE TRAPPING BY MINIMA REACTIVATION

We identify the fundamental limitation of deadzone trapping in ternary quantization as the fact that the dead weights provide no meaningful signal to the model output in the forward pass. This creates a vicious cycle where trapped weights cannot contribute to learning and struggle to escape the deadzone effectively, significantly impeding convergence.

Our core motivation is to allow dead weights to contribute to the forward output, even with minimal but informative values, which can break this trapping by receiving a more direct and informative gradient signal. Therefore, we propose to *repurpose* dead weights to provide informative signals, which enhance model capacity and establish clean gradient pathways.

To implement this, we intuitively propose Minima Reactivation, preserving the sign information of dead weights, reactivating them as distinct values $0_-$ and $0_+$, representing negative and positive minima, respectively, as shown in Fig. 2 (b). This creates a quaternary weight representation $\tilde{w}_i \in$

$\{-1, 0_-, 0_+, +1\}$. Crucially, to isolate the impact of increasing the representation capability from 3 to 4 values while preserving only sign information, we replace multiplication with a constant-magnitude mapping. For any input $x$, the operation yields a value $\pm\varepsilon$ :

$$x \cdot 0_+ = \text{sign}(x)\varepsilon, \quad x \cdot 0_- = -\text{sign}(x)\varepsilon. \tag{4}$$

Formally, let $\tilde{W} = (\tilde{w}_1, \ldots, \tilde{w}_n)$ represent the quaternary weight vector, and define the set of indices in the deadzone as $D = \{i | -\Delta < w_i < \Delta\}$. The forward pass can then be converted to:

$$Y = X\tilde{W}\alpha = \alpha \underbrace{\sum_{i \in \bar{D}} \text{sign}(w_i)x_i}_{\text{original output}} + \varepsilon \underbrace{\sum_{i \in D} \text{sign}(x_i)\text{sign}(w_i)}_{\text{online bias-like term}}, \tag{5}$$

where the $\bar{D}$ denotes the set of indices not in the deadzone. This formulation reveals that previously dead weights now contribute meaningfully to the output through the bias-like term. Consequently, these weights $w_i$ receive informative gradients from backpropagation, denoted as

$$\frac{\partial L}{\partial w_i} = \varepsilon \cdot \text{sign}(x_i) \cdot \frac{\partial L}{\partial Y}, \quad \forall i \in D. \tag{6}$$

Compared to previous naive ternary quantization, where dead weights receive essentially random gradients in Eq. 3, this approach enables these dead weights to contribute a meaningful, non-zero signal that directly influences the final output. This allows them to receive more informative gradient signals proportional to the downstream loss, thereby providing an effective optimization direction. As shown by the GSNR measurements in Fig. 3 *(middle)*, Minima Reactivation provides reactivated weights with higher GSNR, thereby mitigating the dead-zone trapping issue and enabling more efficient optimization.

**Limitations:** While this Minima Reactivation method demonstrates the theoretical viability of deadzone repurposing, we identify two practical limitations: (1) **Noisy gradients and unstable training:** The gradients for reactivated weights in Eq. 3 still rely on the STE for the $\text{sign}(\cdot)$ operation, introducing noise and unstable training, as shown in Fig. 3 *(middle)*, yielding only marginal accuracy gains. (2) **Non-negligible inference overhead:** The additional bias-like term, which is input-dependent, introduces non-negligible inference overhead, as it requires computation for every forward pass.

These insights motivate our final Tequila method, which retains the core concept of deadzone reactivation while introducing key optimizations to overcome these limitations, as detailed in the next subsection.

## 3.2 TEQUILA: REPURPOSING DEAD WEIGHT AS DYNAMIC BIAS

This section introduces Tequila, a trapping-free quantization method that reactivates deadzone-trapped weights to enhance model capacity and restore optimization potential without sacrificing hardware efficiency. Tequila's core innovation lies in repurposing the deadzone from a fundamental limitation into a source of adaptability through the following three key designs.

**Differentiable Reactivation:** To address the noisy gradient and unstable training problem in Minima Reactivation, we replace the non-differentiable mapping to a constant $\pm\varepsilon$ with a scaling for dead weight $w_i$, allowing the computation of reactivation values as $\lambda w_i$, resulting in a smooth, differentiable reactivation function. That is, in forward pass, the Eq. 5 is converted as:

$$Y = X\hat{W}\alpha \approx \alpha \sum_{i \in \bar{D}} \text{sign}(w_i)x_i + \lambda \sum_{i \in D} \text{sign}(x_i)w_i. \tag{7}$$

Crucially, this design bypasses the STE, providing direct and informative gradients that enable effective optimization of previously trapped weights.

**Repurposing Dead Weights as Biases:** To eliminate the non-negligible inference overhead of Minima Reactivation, we repurpose dead weights as actual biases, thereby converting online computation into an offline one, i.e., setting $\lambda \sum_{i \in D} \text{sign}(x_i)w_i \approx \lambda \sum_{i \in D} w_i$ in Eq. 7. This simplification is justified by two reasons. First, it remains faithful to our core motivation of enabling dead

weights to contribute to the output; a bias term achieves this in an activation-free manner, perfectly aligning with our hypothesis while minimizing overhead. Second, we find empirical support for this approximation: the cosine similarity between the original bias-like vector and the simplified bias vector is high ($> 70\%$), as shown in Fig. 4, validating that our simplification retains the essential functional behavior.

**Hybrid Roles of Reactivated Weights:** While converting dead weights to pure biases ($\sum_{i \in D} \lambda w_i$) provides clean gradients, it discards valuable input information. Tequila overcomes this limitation by assigning reactivated weights to hybrid roles. In addition to functioning as an dynamic bias, these weights are simultaneously maintained as participants in the ternary matrix multiplication. This dual role creates a mixed gradient from both the standard ternary pathway and the direct bias pathway. Consequently, the optimization process preserves crucial input information while benefiting from a direct, informative gradient signal, driving more effective training.

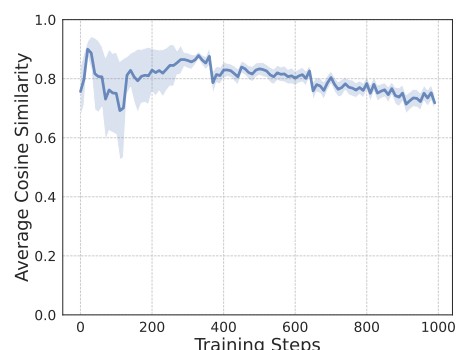

Figure 4: The average cosine similarity of bias-like term in Eq. 7 and pure bias term Eq. 8 during training of Tequila.

With these three key designs, the Tequila forward pass combines efficient ternary operations with dynamic biases:

$$Y = XQ(W) + C(W) = X\hat{W}\alpha + \sum_{i \in D} \lambda w_i, \tag{8}$$

where the bias term $C(W) = \sum_{i \in D} \lambda w_i$ acts as a residual connection for weights within the deadzone. This formulation directly yields superior gradients for these dead weights:

$$\frac{\partial L}{\partial w_i} = x_i \frac{\partial L}{\partial Y} + \lambda \frac{\partial L}{\partial Y}, \quad \forall i \in D, \tag{9}$$

thereby preserving input-dependent information and delivering a direct, informative gradient signal to enable effective optimization. As shown by the GSNR measurements in Fig. 3 *(right)*, Tequila provides high and stable GSNR during the training, thereby enabling more efficient optimization.

**Advantages:** Tequila provides five key advantages over existing ternary quantization methods:

(1) **Enhanced Model Capacity**: Reactivating dead weights effectively expands the model parameter space without increasing computational complexity during inference.

(2) **Trapping-free Optimization**: By providing direct, informative gradients, Tequila enables stable escape from deadzone, achieving trapping-free weight optimization.

(3) **Training Stability**: The differentiable reactivation function ensures stable optimization while maintaining quantization constraints, resulting in more consistent, reliable training convergence.

(4) **Plug-and-play Design**: Tequila is a simple and plug-and-play module that can be easily integrated into most existing ternary quantization methods.

(5) **Nearly Zero Inference Overhead**: The input-agnostic bias term can be precomputed offline and seamlessly fused into the computation kernel, achieving nearly zero inference overhead. This preserves the hardware efficiency of pure ternary quantization.

## 4 EVALUATION

To validate the efficacy of Tequila, we conduct comprehensive experiments evaluating its performance against state-of-the-art ternary quantization methods. All experimental results are averaged on three independent runs with random seeds. In all tables, the best and second-best results are highlighted in **purple** and **blue** color, respectively, and the result of the full-precision method is set to gray color for reference.

| Size | Method | ARC-e | ARC-c | HelS | PIQA | WinG | GPQA | Average |
|------|--------|-------|-------|------|------|------|------|---------|
| 1B | BF16 | 0.654 | 0.313 | 0.477 | 0.742 | 0.603 | 0.222 | 0.502 |
| | LSQ | 0.376 | 0.177 | 0.258 | 0.574 | 0.506 | 0.231 | 0.354 |
| | SEQ | 0.421 | 0.180 | 0.273 | 0.604 | 0.510 | 0.232 | 0.370 |
| | DLT | 0.424 | 0.174 | 0.256 | 0.563 | 0.513 | **0.277** | 0.368 |
| | TWN | 0.407 | 0.220 | 0.284 | 0.601 | 0.492 | 0.212 | 0.363 |
| | AbsMedian | 0.567 | 0.251 | 0.339 | 0.674 | 0.533 | 0.222 | 0.431 |
| | AbsMean | 0.603 | 0.259 | 0.360 | 0.683 | 0.541 | 0.227 | 0.445 |
| | **Tequila+AbsMedian** | **0.645** | **0.308** | **0.393** | **0.719** | **0.558** | **0.237** | **0.477** |
| | **Tequila+AbsMean** | **0.645** | **0.305** | **0.391** | **0.710** | **0.542** | 0.232 | **0.471** |
| 3B | BF16 | 0.745 | 0.422 | 0.552 | 0.768 | 0.691 | 0.303 | 0.580 |
| | LSQ | 0.431 | 0.200 | 0.294 | 0.599 | 0.522 | 0.239 | 0.354 |
| | SEQ | 0.498 | 0.231 | 0.303 | 0.645 | 0.529 | 0.258 | 0.411 |
| | DLT | 0.361 | 0.161 | 0.260 | 0.572 | 0.496 | 0.272 | 0.354 |
| | TWN | 0.692 | **0.351** | 0.462 | 0.734 | 0.586 | 0.237 | 0.510 |
| | AbsMedian | 0.636 | 0.299 | 0.406 | 0.713 | 0.558 | 0.181 | 0.466 |
| | AbsMean | 0.672 | 0.329 | 0.439 | **0.735** | 0.582 | **0.301** | 0.510 |
| | **Tequila+AbsMedian** | **0.701** | 0.345 | **0.449** | 0.733 | **0.611** | 0.237 | **0.513** |
| | **Tequila+AbsMean** | **0.702** | **0.346** | **0.464** | **0.739** | **0.627** | **0.303** | **0.530** |

Table 1: Comparison of Tequila method with different ternary quantization methods

## 4.1 EXPERIMENTAL SETUP

We provide a comprehensive overview of our experimental configuration below, with additional implementation details available in the Appendix E.

**Datasets, Models and Evaluation:** We utilize the LLaMA-3.2-1B, LLaMA-3.2-3B (Touvron et al., 2023) and Qwen3-4B Bai et al. (2023) models as our base architectures, employing a group size of 128 throughout our experiments unless otherwise specified. For quantization-aware training, we use 10B tokens sampled from the UltraFineWeb dataset (Wang et al., 2025c). Following established practices in ternary quantization research (Liu et al., 2025; Chen et al., 2024; Ma et al., 2025), we evaluate model performance with lm-evaluation-harness (Gao et al., 2024) on five zero-shot benchmarks: PIQA (Bisk et al., 2020), ARC-Easy/Challenge (ARC-e/ARC-c) (Clark et al., 2018), HellaSwag (HelS) (Zellers et al., 2019), GPQA-Diamond (Rein et al., 2023) and Wino-Grande(WinG) (Sakaguchi et al., 2021). Details for benchmarks are in Appendix E.1.

**Baselines:** We compare Tequila against several quantization method baselines, which represent the methods used in existing state-of-the-art (SOTA) ternary LLMs. These include two types of quantization methods: (1) *static methods*: TWN Li et al. (2016), AbsMedian and AbsMean used in BitNet (Ma et al., 2025; Wang et al., 2023), Spectra (Kaushal et al., 2025), and BitCPM (Team et al., 2025); and (2) *learnable methods*: DLT in TernaryLLM (Chen et al., 2024), LSQ (Esser et al., 2019), and SEQ used in ParetoQ (Liu et al., 2025). In addition to comparing quantization methods, we also directly compare against those well-trained ternary LLMs. Further discussion about the baselines is in Appendix E.4.

**Implementation Details:** All experiments are conducted on 16 GPUs for training, with inference performance evaluated on an Intel 8263C CPU. Following established practices (Liu et al., 2025), we quantize all linear layers within the transformer architecture. The sequence length for input and output is 1024. The learning rate is set as a fixed value of $10^{-4}$. Given that Tequila is designed as a plug-and-play solution, the AbsMean in Eq. 2 was selected for Tequila's base quantization method due to its prevalence in open-source ternary large language models. We set $\lambda = 10^{-3}$ for Tequila by default. The LLM trained by Tequila is called **TequilaLLM**.

## 4.2 PERFORAMCNE EVALUATION

**Comparison of Different Ternary Quantization Methods:** To evaluate the effectiveness of Tequila, we conduct QAT with different ternary quantization methods with 10B tokens and evaluate their performance. Our Tequila is plugged into the AbsMedian and AbsMean quantization

| Model | Size | #Tokens | ARC-e | ARC-c | HelS | PIQA | WinG | Average |
|---|---|---|---|---|---|---|---|---|
| LLaMA3.2 | 1B | - | 0.654 | 0.313 | 0.477 | 0.742 | 0.603 | 0.558 |
| TernaryLLM* | 1B | 10B | 0.424 | 0.174 | 0.256 | 0.563 | 0.513 | 0.386 |
| ParetoQ* | 1B | 10B | 0.421 | 0.180 | 0.273 | 0.604 | 0.510 | 0.398 |
| LLM-QAT | 1B | 100B | 0.360 | **0.262** | 0.313 | 0.551 | 0.496 | 0.397 |
| BitNet | 1.3B | 100B | **0.549** | 0.242 | 0.377 | 0.688 | **0.558** | 0.483 |
| Spectra | 1.1B | 100B | 0.563 | 0.246 | **0.388** | **0.693** | **0.555** | **0.489** |
| **TequilaLLM** | 1B | 10B | **0.645** | **0.305** | **0.391** | **0.710** | 0.542 | **0.519** |
| LLaMA3.2 | 3B | - | 0.745 | 0.422 | 0.552 | 0.768 | 0.691 | 0.636 |
| TernaryLLM* | 3B | 10B | 0.361 | 0.161 | 0.260 | 0.572 | 0.496 | 0.370 |
| ParetoQ* | 3B | 10B | 0.498 | 0.231 | 0.303 | 0.645 | 0.529 | 0.441 |
| LLM-QAT | 3B | 100B | 0.445 | 0.307 | 0.434 | 0.627 | 0.506 | 0.464 |
| BitNet | 3B | 100B | 0.614 | 0.283 | 0.429 | 0.715 | 0.593 | 0.527 |
| Spectra | 3.9B | 100B | **0.660** | 0.319 | **0.483** | **0.744** | **0.631** | **0.567** |
| **TequilaLLM** | 3B | 10B | **0.702** | **0.346** | 0.464 | 0.739 | 0.627 | **0.576** |

Table 2: Comparison of TequilaLLM with other ternary LLMs across different model sizes and training token counts (#Tokens); * indicates LLMs obtained from our reproduction.

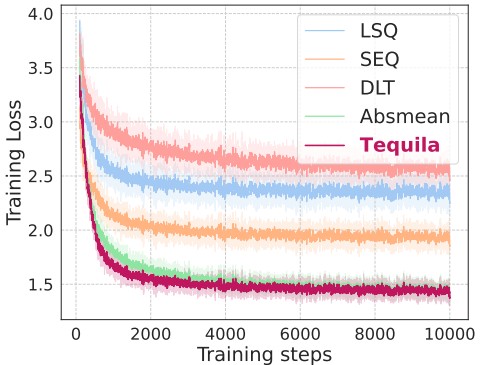

Figure 5: Evaluation of Tequila on convergence speed compared to SOTA ternary quantization.

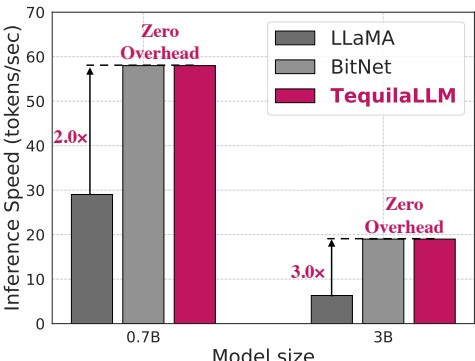

Figure 6: Inference speed of TequilaLLM versus BF16 LLaMA and ternary BitNet.

method. The experimental results in Table 1 show that Tequila outperforms all baselines on both 1B and 3B models, achieving an average accuracy gain of $> 2.6\%$ over SOTA methods. Specifically, on both ARC-Easy and ARC-Challenge benchmarks, Tequila achieves significant $> 4\%$ accuracy gains over SOTA methods, while matching the BF16 performance with only a minimal gap ($< 1\%$).

An important observation is that learnable ternary quantization methods generally underperform static ones. We attribute this to the fact that increasing learnable parameters slows convergence and makes optimization more prone to getting stuck in local optima. This aligns with the broader trend of using static AbsMean quantization method in open-source ternary LLMs (Liu et al., 2025; Chen et al., 2024; Ma et al., 2025), confirming our design decision.

**Comparison with Different Ternary LLMs:** To further evaluate the effectiveness of Tequila, we name the resulting model trained by Tequila as Tequila and compare it against existing ternary LLaMA-based LLMs. We reproduce methods Liu et al. (2025); Chen et al. (2024) with available training code and train them on 10B tokens from the UltraFineWeb dataset using identical hyperparameters for a fair comparison. For models without available implementations, we report results from their original papers or published weights. The GPQA benchmark is excluded from comparison as it is not consistently reported across baselines. As shown in Table 2, TequilaLLM achieves the best average accuracy in the benchmarks. Remarkably, Tequila achieves superior performance using significantly fewer training tokens than other well-trained ternary LLMs, demonstrating both faster convergence and higher final accuracy. Specifically, our TequilaLLM-3B model outperforms the SOTA ternary LLM, Spectra-3.9B, by 0.9% in average accuracy while using only 10% of the training tokens. These results robustly validate the effectiveness of Tequila's adaptive reactivation strategy in resolving the deadzone trapping problem.

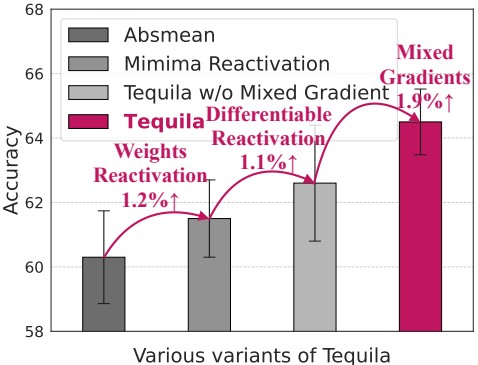

Figure 7: Ablation study comparing Tequila against its variants.

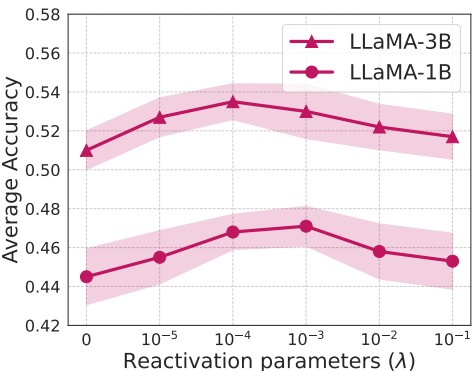

Figure 8: Sensitivity evaluation for the reactivation parameter $\lambda$.

**Convergence Analysis:** To demonstrate that Tequila's more direct and informative gradients enable faster information recovery, we compare the training loss convergence of a 1B model using Tequila against ternary baselines over 10,000 steps. As shown in Fig. 5, Tequila achieves a faster convergence rate than other baselines. This result validates the effectiveness of our core innovation: reactivating dead weights as adaptive biases by a differentiable reactivation function can obtain superior gradient signals, thereby enhancing optimization.

**Inference Efficiency:** Theoretically, Tequila introduces nearly zero inference overhead, as the reactivation bias terms are input-independent and can be precomputed offline. The additional computational cost for bias addition is negligible, measured at less than 0.1%. To empirically validate this claim, we evaluate the token generation speed of Tequila against BitNet and a BF16 LLaMA baseline on an Intel 8263C CPU. Due to limitations in the compiled BitNet model, we conduct experiments using 0.7B and 3B model sizes. We accelerate both Tequila and BitNet using the efficient lookup table paradigm (Wang et al., 2025b) to eliminate multiplication operations, as shown in Fig. 9. The results in Fig. 6 show that TequilaLLM maintains a $3.0\times$ speedup over the LLaMA-3.2, matching the practical inference speed of BitNet, demonstrating that Tequila introduces nearly zero overhead compared to pure ternary methods.

**Ablation Study:** To analyze the individual contributions of Tequila's components in addressing deadzone trapping, we conduct an ablation study on a 1B model using the ARC-Easy benchmark. We attribute Tequila's benefits to three key aspects: the reactivated forward signal, differentiable reactivation, and the hybrid roles of reactivated weights. To evaluate these aspects, we compare the following variants of Tequila: (1) *AbsMean:* This baseline disables all reactivation aspects to evaluate Tequila's overall effectiveness. (2) *Minima Reactivation:* This variant reactivates dead weights in-place as signed minima (Sec. 3.1), enabling the reactivated forward signal but still relying on the STE for gradients. (3) *Tequila w/o Mixed Gradients:* This variant treats dead weights as biases only by differentiable reactivation. It replaces the gradients in Eq. 9 with $\frac{\partial L}{\partial w_i} = \lambda \frac{\partial L}{\partial Y}$ for $\forall i \in D$, while keeping the forward pass unchanged.

The results in Fig. 7 demonstrate the incremental effectiveness of each aspect. First, *Minima Reactivation* outperforms the *AbsMean*, confirming that reactivating dead weights enhances model capacity. Second, *Tequila w/o Mixed Gradients* surpasses *Minima Reactivation*, demonstrating that differentiable reactivation is more effective than the STE, as it provides direct backpropagation to mitigate deadzone trapping. Finally, the *Tequila* achieves superior performance to *Tequila w/o Mixed Gradients*, validating that mixed gradients from the residual pathway are more effective than the single gradient from an input-agnostic bias. This shows that assigning dead weights a hybrid role (functioning as both weights and biases) is more suitable than a pure bias assignment. This ablation study conclusively demonstrates the individual and combined importance of the reactivated forward signal, differentiable reactivation, and hybrid roles of reactivated weights.

**Impact of the Reactivation Parameter $\lambda$:** The choice of the reactivation parameter $\lambda$ is critical for Tequila. An excessively high value of $\lambda$ may cause the bias to dominate the output, while a value

| Model | ARC-e | ARC-c | HelS | PIQA | WinG | Average |
|---|---|---|---|---|---|---|
| BF16 | 0.8186 | 0.5213 | 0.5423 | 0.7769 | 0.6772 | 0.6673 |
| AWQ | 0.2702 | 0.2244 | 0.2587 | 0.5305 | 0.5122 | 0.3592 |
| AbsMean | 0.6915 | 0.3507 | 0.4616 | 0.7339 | 0.5856 | 0.5647 |
| **Tequila** | **0.7538** | **0.4189** | **0.4681** | **0.7383** | **0.6133** | **0.5985** |

Table 4: Performance comparison (accuracy) for Qwen3-4B.

that is too low renders the reactivation ineffective; if $\lambda = 0$, Tequila degenerates to standard ternary quantization. To analyze the sensitivity of $\lambda$, we evaluate average accuracy on five benchmarks across a range of values: $\lambda \in \{0, 10^{-5}, 10^{-4}, 10^{-3}, 10^{-2}, 10^{-1}\}$. The results in Figure 8 indicate that even a small $\lambda$ provides a noticeable gain, and performance is robust across a wide range of values. This suggests the model can effectively adapt the dead weights into useful biases during training, converging to a near-optimal configuration regardless of the value of $\lambda$. And the optimal $\lambda$ is correlated with model size, with larger models preferring a smaller $\lambda$.

**The Impact of Quantization Granularity:** Quantization granularity presents a fundamental trade-off between a model's efficiency and its performance. Per-token quantization enables high acceleration but introduces significant quantization error. Conversely, per-group quantization mitigates this error at the cost of reduced efficiency, due to the overhead of storing and applying scaling matrices. We evaluate Tequila across various granularities: per-token, per-channel, and per-group with group size 128. The results in Table 3 indicate that Tequila exhibits minimal performance loss across different granularities. This robustness stems from its ability to use reactivated biases to compensate for quantization errors.

| Tequila | Average Acc |
|---|---|
| per-tensor | 0.463 |
| per-channel | 0.471 |
| per-group | 0.471 |

Table 3: Average accuracy of Tequila across quantization granularities on a 1B Model

**Experiments on Qwen3-4B:** To validate the scalability and generalization of Tequila, we extend our evaluation to the Qwen3-4B model Bai et al. (2023). We compare Tequila against AbsMean and a standard Post-Training Quantization (PTQ) approach using AWQ Lin et al. (2023). The results is shown in Table 4. The AWQ suffers a catastrophic performance drop, which underscores the necessity of QAT for aggressive ternary quantization in LLMs. Tequila consistently outperforms other baselines across all tasks, confirming that our proposed reactivation mechanism delivers a consistent advantage by providing more informative gradients and mitigating the dead zone problem, even at the 4B scale.

## 5 CONCLUSION

In this paper, we first identified **deadzone trapping** as a fundamental obstacle to efficient and accurate ternary quantization of large language models for on-device deployment. Deadzone trapping, where weights become trapped in ineffective oscillation around the quantization boundary due to less informative gradients, severely diminishes model capacity and impedes optimization.

To overcome this challenge, we introduced Tequila, a novel trapping-free ternary quantization method. Tequila repurposes trapped weights as adaptive dynamic biases, successfully reactivating them to enhance model expressiveness with nearly zero inference overhead. Crucially, this approach provides direct gradient signals, enabling efficient escape from the deadzone and substantially accelerating quantization-aware training. Extensive evaluations on five benchmarks demonstrate that Tequila outperforms state-of-the-art ternary methods, closing the gap to full-precision performance while using limited training data, and preserving the computational benefits of ternary quantization, delivering up to $3\times$ inference speedup.

Looking forward, Tequila establishes a new direction for efficient model compression. **The concept of dynamically repurposing dead weights opens promising avenues for future research into extreme quantization.** We believe our work contributes a practical and scalable path toward bringing advanced LLM capabilities to resource-constrained devices.

## 6 ACKNOWLEDGEMENTS

The paper is partially supported by Hong Kong Research Grants Council grants #11205424, #11203523, and C1042-23GF.

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

# Appendix

## A    THE USAGE OF LLMs

We state that the LLM is used exclusively for polishing the English text and grammar in this manuscript, prompted with: "Please polish and rephrase the following sentences: #input". All technical content, ideas, methodologies, experimental results, analyses, and conclusions are the original work of the authors. The LLM acted solely as a writing assistant and did not contribute to the intellectual substance of the research.

## B    RELATED WORK

### B.1    GENERAL QUANTIZATION

Quantization (Dettmers et al., 2021; 2022; Lin et al., 2023; Frantar et al., 2022) is a well-established technique for improving the efficiency of Large Language Models (LLMs) by reducing the precision of weights and activations. However, low-precision quantization (Lin et al., 2023; Frantar et al., 2022) often leads to mixed-precision matrix multiplication, where weights and activations have different data types. This requires specific hardware support for efficient computation, which is a significant limitation for edge and mobile deployment, given the extreme diversity of devices in these environments.

While activation-weight quantization methods (Dettmers et al., 2022; Xiao et al., 2023; Huang & Wu, 2025) attempt to mitigate this by using a unified low-precision format for both weights and activations, they still face challenges. These methods often require specialized hardware adaptation and suffer from high quantization error in activations due to outlier issues (Xiao et al., 2023), preventing activations from reaching the same effective precision as weights. Consequently, existing general quantization methods struggle to enable efficient LLM deployment on diverse edge and mobile platforms.

### B.2    TERNARY QUANTIZATION

Ternary quantization Li et al. (2016); Zhu et al. (2016), or 1.58bit quantization. presents a compelling alternative by constraining weights to ternary values, typically $\{-1, 0, +1\}$. Beyond the substantial memory savings, this approach transforms the core matrix multiplication operation into efficient addition operations by replacing multiplications with conditional sign flips. This intrinsic hardware-friendliness makes it particularly suitable for resource-constrained edge and mobile hardware.

Early research on ternary quantization Li et al. (2016); Zhu et al. (2016); Leng et al. (2018) primarily focused on refining the quantization function, particularly the selection of threshold parameters and scaling factors. The foundational Ternary Weight Networks (TWN) (Li et al., 2016) assumed a Gaussian weight distribution to determine optimal thresholds that minimize the distortion between full-precision and quantized weights. Trained Ternary Quantification (TTQ) (Zhu et al., 2016) advanced this paradigm by introducing trainable scaling factors, enabling models to learn optimal ternary representations directly during training. Further extending these ideas, Leng et al. (2018) formulated the problem using the Alternating Direction Method of Multipliers (ADMM) to iteratively optimize both scaling factors and thresholds.

With the emergence of Large Language Models (LLMs) Wu et al. (2023); Floridi & Chiriatti (2020); Zhang et al. (2022), the limitations of early ternary quantization methods have become apparent. The generative capabilities of LLMs are highly sensitive to precision loss, which often leads to unacceptable performance degradation under existing ternary techniques. This challenge has spurred two primary research directions. One line of work employs Post-Training Quantization (PTQ) techniques Lin et al. (2023); Frantar et al. (2022) to mitigate performance loss efficiently Xiao et al. (2025). However, concerns over PTQ's potential performance drop have led the most of work to explore Quantization-Aware Training (QAT) Chen et al. (2025) for more robust recovery  Within QAT-based ternary methods, a divergence in strategy exists: some approaches, such as those in the

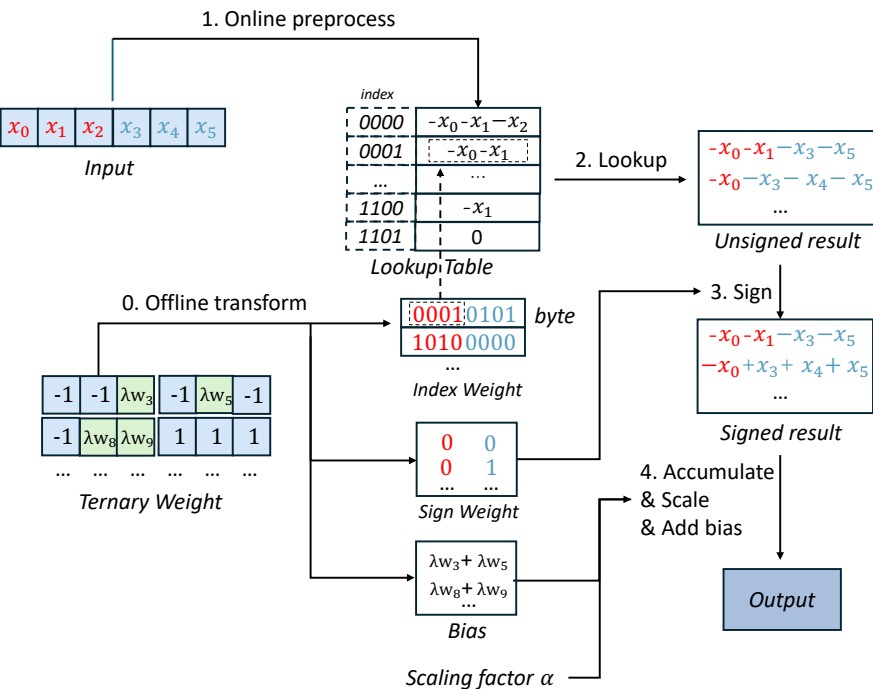

Figure 9: The Lookup Table-Based Inference designed for Tequila. Weights are packed into indices, signs, and a precomputed bias offline. At runtime, inputs are processed to construct a lookup table containing various unsigned intermediate values. The system then uses the weight indices to retrieve these values from the table. After that, the 1-bit sign values are applied to generate signed results and are accumulated, and the per-channel bias is added to produce the final output.

BitNet family Ma et al. (2025); Wang et al. (2023; 2025b), utilize the straightforward AbsMean quantization, while others, like Chen et al. (2024) and Liu et al. (2025), have adapted more sophisticated techniques such as Learned Step Size Quantization (LSQ) Esser et al. (2019) to the ternary setting.

Despite these advancements, these existing ternary LLMs still suffer from the deadzone trapping issue, where a large number of weights become trapped in a cycle of ineffective oscillation around the deadzone boundary, severely impeding model capacity and convergence. Therefore, we propose Tequila to achieve trapping-free quantization.

## C  QUANTIZATION-AWARE TRAINING

Quantization-Aware Training (QAT) Chen et al. (2025); Liu et al. (2025) is employed to recover model performance by incorporating quantization into the training loop. The core idea of QAT is to model the effects of quantization during the forward pass, allowing the model to adapt its parameters to the subsequent precision loss. This is achieved through a *fake quantization* process, where full-precision weights and activations are passed through a quantization function $Q(\cdot)$ that mimics the behavior of integer arithmetic. Critically, the backward pass leverages the Straight-Through Estimator (STE) Yin et al. (2019) to approximate gradients through this otherwise non-differentiable operation.

The standard QAT procedure can be summarized as follows:

(1) Maintain a full-precision weight copy $W$ as the master weights for accumulation.

(2) During the forward pass, generate quantized weights $\hat{W} = Q(W)$ and scaling factor $\alpha$ for computation.

(3) Compute the output $Y$ using the quantized weights $\hat{W} = Q(W)$, scaling factor $\alpha$ and activations $X$.

(4) Compute the Loss $L$ using output $Y$ and begin backward pass.

(5) During the backward pass, full-precision gradients $\frac{\partial L}{\partial W}$ are computed with the STE, the gradients are passed directly to the full-precision weights: $\frac{\partial L}{\partial W} \approx \frac{\partial L}{\partial \hat{W}}$.

(6) Update the full-precision weights $W$ using the approximated gradients via a standard optimizer (e.g., SGD, Adam).

By maintaining and updating full-precision master weights, QAT provides a stable optimization process while training a model that is robust to the quantization errors it will encounter during inference. This makes it the preferred method for achieving high accuracy with ultra-low precision, such as ternary or binary quantization.

## D  INFERENCE DESIGN

Tequila seamlessly integrates with the lookup table paradigm to enable efficient, multiplication-free inference. As illustrated in Figure 9, our system operates in two phases: an offline packing stage and an efficient online inference stage.

During the offline phase, the ternary weights and reactivation biases are packed into compact data structures, including index weights, sign weights, and channel-wise biases. To maximize efficiency, every three weights are packed into a 4-bit index and a 1-bit sign value.

At inference time, the input values are preprocessed into a lookup table within segments. For each segment, the corresponding weight index is used to retrieve the results from the lookup table. This process entirely replaces multiplication operations with efficient table lookups. The retrieved result is then combined with the sign weight to determine the final polarity, and subsequent accumulation across segments is followed by the addition of the channel-wise bias term. This results in a highly optimized inference path that maintains the theoretical hardware efficiency of ternary quantization while requiring only minimal modifications to existing inference frameworks.

## E  MORE EXPERIMENTAL DETAILS

### E.1  EVALUTAION BENCHMARKS

The evaluation of language models has evolved beyond simple word prediction to assessing their ability to understand and apply knowledge in a human-like manner. This requires benchmarks that probe deeper cognitive capabilities, such as commonsense reasoning, logical deduction, and specialized knowledge. This section introduces a suite of prominent benchmarks designed for this purpose: PIQA, ARC-Easy, ARC-Challenge, HellaSwag, GPQA, and WinoGrande.

PIQA (Physical Interaction Question Answering) Bisk et al. (2020) focuses on physical commonsense reasoning, testing a model's understanding of how the everyday physical world works. The benchmark presents questions about the mechanics of physical actions (e.g., "How do you stabilize a wobbly table?") and requires choosing the correct solution from two options. Success on PIQA indicates that a model possesses a foundational knowledge of physical laws and object interactions.

The ARC (AI2 Reasoning Challenge) dataset  Clark et al. (2018) is divided into two tiers to assess scientific knowledge and reasoning. The ARC-Easy set contains grade-school-level science questions that are often answerable through simple fact retrieval. In contrast, the ARC-Challenge set is specifically curated to be difficult, consisting of questions that require complex reasoning and a deeper understanding of scientific concepts, posing a significant challenge for even advanced models.

HellaSwag  Zellers et al. (2019) is a benchmark for evaluating contextual commonsense reasoning. It presents a beginning of a situation (e.g., "A person is folding a paper towel") and challenges the model to select the most plausible continuation from four options. The distractors are generated by adversarial models, making them deceptively plausible and ensuring the task cannot be solved by simple word association, but rather requires a nuanced understanding of event dynamics.

GPQA Rein et al. (2023) represents a significant leap in difficulty, designed as a "graduate-level" benchmark for highly specialized knowledge. The questions, written by domain experts in biology,

physics, and chemistry, are exceptionally challenging and "Google-proof," meaning they are difficult to answer by simply searching the web. On GPQA-Diamond, the set of GPQA's 198 most difficult questions, PhD experts achieve 65% accuracy, while skilled non-experts with web access only reach 34%.

WinoGrande Sakaguchi et al. (2021) is a large-scale dataset for assessing commonsense reasoning through pronoun resolution. Inspired by the Winograd Schema Challenge, it presents sentences with ambiguous pronouns (e.g., "The trophy didn't fit in the suitcase because it was too big.") and requires the model to determine the referent of "it." WinoGrande is designed to be adversarial, with a focus on reducing spurious statistical biases present in earlier datasets, forcing models to rely on genuine commonsense understanding.

Together, these benchmarks provide a multifaceted evaluation framework, testing language models on everything from everyday physical intuition (PIQA, HellaSwag) and general scientific knowledge (ARC) to expert-level understanding (GPQA) and nuanced linguistic reasoning (WinoGrande).

### E.2 GRADIENT SIGNAL-TO-NOISE RATIO (GSNR) MEASUREMENT

The Gradient Signal-to-Noise Ratio (GSNR) Liu et al. (2020) is a crucial metric for quantifying the reliability of gradient updates for model weights. It is defined as the ratio of the squared expectation to the variance of a weight's gradients over the data distribution. Formally, for a weight $w_i$, its GSNR is calculated as:

$$\text{GSNR}(w_i) = \frac{(\mathbb{E}_{\mathbf{x}\sim\mathcal{D}}[\nabla_{w_i}L])^2}{\text{Var}_{\mathbf{x}\sim\mathcal{D}}[\nabla_{w_i}L]}, \tag{10}$$

where $\nabla_{w_i}L$ is the gradient of the loss with respect to $w_i$ for a data sample $\mathbf{x}$. A high GSNR indicates a stable and consistent gradient signal across the dataset, whereas a low GSNR suggests noisy, uninformative updates that hinder effective optimization.

To evaluate the impact of our method on , we measure the GSNR specifically for dead weights, active weight and reactive weights that are trained by AbsMean, Minima Reactivation and Tequila. We estimate the expectation and variance in Eq. 10 using a 1024 training data samples and compute the average GSNR. This comparative analysis, shown in Figure 3, directly demonstrates that our reactivation mechanism provides a clearer optimization signal by significantly elevating the GSNR of previously dead weights.

### E.3 GENERALIZABILITY TO OTHER QUANTIZATION SCHEMES

This paper primarily addresses the *deadzone trapping* problem, a fundamental challenge in *ternary* quantization that arises from uniquely fixing the zero point to eliminate multiplications. In higher-bit regimes (e.g., 2-bit, 4-bit), which are multiplication-based, this specific issue is effectively mitigated through techniques like zero-point shifting Liu et al. (2025). Consequently, the primary contribution of our work is the identification and resolution of a problem inherent to the ternary paradigm.

However, the core innovation of Tequila that repurposes a subset of weights to form a dynamic bias term is a general mechanism to enhance model capacity and gradient quality. To validate Tequila's generalizability beyond the ternary setting, we adapt Tequila to the following higher bit schemes.

**Low-bit Quantization Scheme Without Deadzone.** To show the benefit of the weight repurposing method in a non-deadzone quantization scheme, we adapt Tequila into a 2-bit scheme using the levels $\{-1, -0.5, +0.5, +1\}$ (SEQ) by introducing a repurposed-bias term, modifying the forward pass to $Y = XW + \lambda \sum_i W_i$. Results in Table 5, confirm the broader applicability of our Tequila. The integration of the repurposed-bias term consistently improves performance across most tasks, yielding a noticeable 1.04% gain in the average accuracy. This demonstrates that the benefits of Tequila can be effectively translated to other low-precision formats.

**High-bit Quantization Scheme With Deadzone.** We further evaluate Tequila's generalization on high-bit quantization scheme by applying it to INT4 LSQ quantization. As shown in Figure 11 (left), the deadzone trapping effect in INT4 LSQ is less severe than in ternary quantization (Figure 1). Despite this, Tequila effectively mitigates the remaining trapping, leading to a more stable

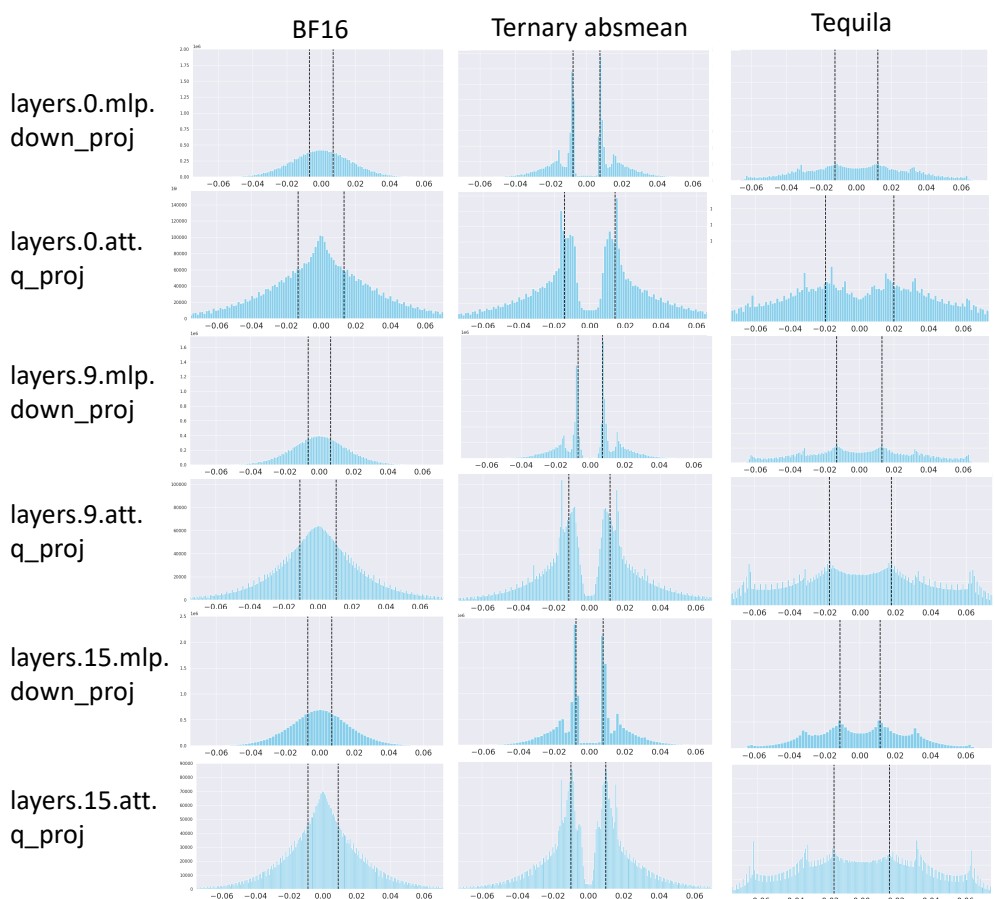

Figure 10: The distribution of weights from randomly selected linear layers in transformers with full-precision (BF16), ternary AbsMean, and our Tequila. The weights from the ternary AbsMean method are trapped at the deadzone boundary, suffering from deadzone trapping. Our Tequila can significantly address this issue. This observation aligns with our findings and demonstrates the effectiveness of Tequila.

weight distribution Figure 11 *(right)*. The resulting performance gains, reported in Table 5, are consistent but modest. This is because the INT4 LSQ baseline is already highly competitive with the BF16 baseline, leaving little room for significant improvement. These results confirm that Tequila generalizes to higher-bit quantization, even when absolute gains are limited by a strong baseline.

### E.4 TERNARY QUANTIZATION BASELINES

Recall the general form of the ternary quantization: Given a full-precision weight vector $W = (w_1, \ldots, w_n)$, the general form of the ternary quantization function $Q(\cdot)$ is defined as:

$$Q(W) = \hat{W}\alpha, \quad \hat{w}_i = \begin{cases} +1, & \text{if } w_i \geq \Delta \\ 0, & \text{if } |w_i| < \Delta \\ -1, & \text{if } w_i \leq -\Delta \end{cases} \tag{11}$$

where $\hat{W} = (\hat{w}_1, \ldots, \hat{w}_n)$ is ternary weights, $\alpha$ is a scaling factor and $\Delta$ is a threshold parameter. Due to the non-differentiable function of $Q(\cdot)$, the gradients for $W$ are approximated using the Straight-Through Estimator (STE) (Zhu et al., 2016; Chen et al., 2024), leading to the following forward pass and backpropagation with input vector $X = (x_1, \ldots, x_n)$:

$$Y = X^T Q(W) = X^T \hat{W}\alpha, \quad \frac{\partial L}{\partial w_i} = \begin{cases} \frac{\partial L}{\partial Y} x_i \alpha, & \text{if } |w_i| \geq \Delta \\ \frac{\partial L}{\partial Y} x_i, & \text{if } |w_i| < \Delta \end{cases}, \tag{12}$$

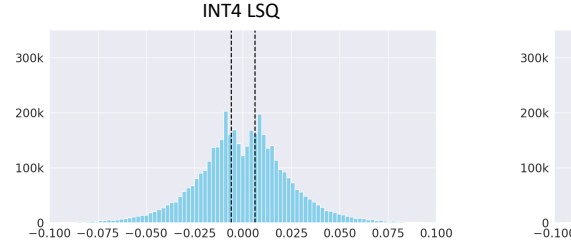
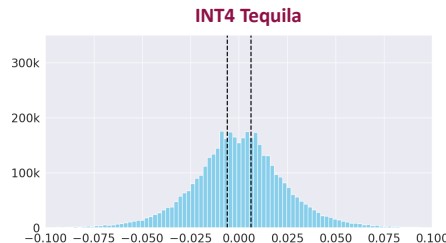

Figure 11: The weight distribution of INT4 *(left)* LSQ and *(right)* Tequila

| Method | Bits | ARC-e | ARC-c | HelS | PIQA | WinG | Average |
|---|---|---|---|---|---|---|---|
| LLaMA-1B | 16 | 0.654 | 0.313 | 0.477 | 0.742 | 0.603 | 0.557 |
| SEQ | 2 | 0.591 | 0.245 | **0.365** | 0.671 | **0.531** | 0.481 |
| **Tequila + SEQ** | | **0.604** | **0.258** | 0.363 | **0.703** | 0.528 | **0.491** |
| LSQ | 4 | 0.670 | 0.310 | 0.458 | 0.741 | 0.587 | 0.553 |
| **Tequila + LSQ** | | **0.672** | **0.313** | **0.459** | **0.747** | **0.593** | **0.557** |

Table 5: Performance comparison of the higher bit scheme with and without the Tequila.

where $L$ denotes the loss of the model prediction.

Previous methods for optimizing ternary quantization can be broadly categorized into two approaches: (1) reducing quantization error, and (2) enhancing the model's expressive capacity.

### E.4.1 REDUCING QUANTIZATION ERROR

A primary line of work forcus on optimizing the threshold $\Delta$ and scaling factor $\alpha$ to reduce the quantization error. It is exemplified by estimation-based methods like Ternary Weight Networks (TWN) (Li et al., 2016), which aim to minimize the reconstruction error between full-precision and quantized weights. This is formalized by the objective:

$$\min_{\Delta,\alpha} |W - \alpha\hat{W}|^2, \tag{13}$$

which has a closed-form solution for $\alpha$ given a fixed threshold $\Delta$:

$$\alpha^* = \frac{1}{|\hat{D}|} \sum_{i \in \hat{D}} |w_i|, \tag{14}$$

where $\hat{D}$ is the set of weights whose absolute value exceeds $\Delta$. However, finding the optimal threshold $\Delta^*$ that minimizes the overall objective is challenging. To circumvent this, TWN hypothesizes that the weights follow a standard Gaussian distribution, leading to the approximation $\Delta^* \approx 0.7 \cdot \mathbb{E}[|W|]$.

This Gaussian assumption often does not hold in modern deep learning models, particularly in LLM, where weight distributions can be highly non-Gaussian. Consequently, the estimated $\alpha$ becomes biased, degrading performance. To address this, subsequent methods like Learned Step Size Quantization (LSQ) (Esser et al., 2019) and DLT (Chen et al., 2024) propose making the scaling factor $\alpha$ a trainable parameter, while typically retaining the heuristic estimation for $\Delta$.

However, optimizing $\alpha$ and $\Delta$ alone cannot resolve the fundamental issue of deadzone trapping. Weights within the deadzone $(-\Delta, \Delta)$ remain long-term pruned during the forward pass and continue to receive only noisy, less informative gradients via the Straight-Through Estimator (STE), which prevents effective recovery. Moreover, in the context of LLMs, simply introducing additional trainable parameters (e.g., for $\alpha$) increases optimization complexity and can make the model more susceptible to converging to poor local optima. As our results in Fig. 5 demonstrate, these methods exhibit significantly slower convergence and higher final loss compared to our approach, underscoring their inherent limitations.

### E.5 Enhancing the Model's Expressive Capacity

Previous work has attempted to enhance model capacity by incorporating bias terms. For instance, DLT Li et al. (2016) introduces a learnable bias during dequantization:

$$Q(W) = \hat{W}\alpha + b, \quad Y = X(\hat{W}\alpha + b) = X\hat{W}\alpha + Xb, \tag{15}$$

where $b$ is a learnable bias. However, this approach breaks the computational efficiency of ternary quantization by introducing the dense full-precision scaling $Xb$. Similarly, SEQ Liu et al. (2025) introduces a bias by reassigning the zero point to a non-zero value $\alpha b$:

$$\hat{w}_i = \begin{cases} +1, & \text{if } w_i \geq \Delta \\ \alpha b, & \text{if } |w_i| < \Delta \\ -1, & \text{if } w_i \leq -\Delta \end{cases} \tag{16}$$

This also destroys efficiency, as the resulting operations are no longer multiplication-free.

While these methods may potentially mitigate the deadzone trapping issue by reactivating the dead-zone, they fundamentally break the hardware efficiency of ternary quantization. In contrast, our method, Tequila, introduces an input-agnostic bias that is precomputed offline. This design directly addresses the deadzone trapping issue while perfectly preserving the computational efficiency of ternary quantization.

