# OpenReview forum: "Tequila: Trapping-free Ternary Quantization for Large Language Models"
_ICLR.cc/2026/Conference — ICLR 2026 Poster_

### Official Review · Reviewer_N2uB · 2025-10-24

**Soundness:** 3
**Presentation:** 3
**Contribution:** 3
**Rating:** 6
**Confidence:** 5

**Summary:**

This paper proposes Tequila, a training-time method to mitigate the “deadzone trapping” problem inherent in ternary quantization of large language models (LLMs). The authors argue that a large number of weights are sparsified to 0 (the “deadzone”) and receive only noisy straight-through estimator (STE) gradients, which leads to ineffective updates and loss of model capacity. Tequila addresses this by reinterpreting these dead weights as dynamic biases, enabling stronger gradients and providing forward signal restoration. The method is nearly overhead-free during inference and is evaluated across multiple benchmarks (ARC, PIQA, HellaSwag, etc.) using LLaMA-based models, outperforming previous state-of-the-art ternary quantization approaches with significantly fewer training tokens and maintaining ternary inference efficiency.

**Strengths:**

1. The paper presents a novel and intuitive perspective into the deadzone problem in ternary quantization and proposes a clean and effective approach (Tequila) to solve this challenge.

2. The proposed method is practical and hardware-friendly, as it maintains the computational advantages of ternary quantization and requires negligible inference-time complexity.

3. Convincing empirical results across multiple benchmarks and model sizes (1B and 3B LLaMA) show significant and consistent improvements over both static and learnable ternary quantization baselines, including BitNet and Spectra.

4. The paper is well-written, with clear figures and mathematical exposition. The insight into trapped weights and how reinterpreting them as biases yields optimization benefits is well motivated.

**Weaknesses:**

1. Tequila risks being a niche solution as this method is tightly coupled with ternary quantization, where the 0 weight level — the "deadzone" — is prominent. It is unclear how or whether Tequila can be extended to other quantization schemes, especially 2-bit symmetric quantization (4-value) or binary quantization (2-value), where such a deadzone may not exist.

2. Since the current mechanism focuses only on reactivating dead weights (quantized to 0), it is not evident whether a similar strategy can be applied to weights that are quantized to ±1 in ternary quantization, or other levels in higher-bit setups.

**Questions:**

1. Can Tequila be extended to 2-bit or 4-bit quantization scenarios where the weight values are {−1, +1} or {−2, −1, +1, +2} and there is no explicit “deadzone”? If not, can you provide a justification or limitations on generalization?

2. Have the authors considered adding similar learnable correction biases globally or per-group even to non-zero weights in ternary quantization?

---

> ### Author Response · Authors · 2025-11-25
> **Response by Authors**
>
> Thanks for the reviewer N2uB for valuable reviews and we feel happy that they find **our work is novel, practical, and well-written.** We will address each concern and update the manuscript accordingly.
>
> > **W1 & Q1:** It is unclear how or whether Tequila can be extended to other quantization schemes, especially 2-bit symmetric quantization (4-value) or binary quantization (2-value), where such a deadzone may not exist. Can Tequila be extended to 2-bit or 4-bit quantization scenarios where the weight values are {−1, +1} or {−2, −1, +1, +2} and there is no explicit “deadzone”?
>
> **A:** The methodology of Tequila that repurposing weights to provide a direct gradient signal is generalizable. While the specific "deadzone trapping" problem is unique to ternary/zero-containing quantizers, the mechanism of repurposing weights as bias can enhance model capacity and optimization in other schemes.
> To validate this, we applied our concept to 2-bit quantization SEQ (-1,-0.5,+0.5,+1) by repurposing all weights as bias, modifying the forward pass to $Y = XW + \lambda \sum_iW_i$, which As shown in the results below (and also in Table 5), adding a repurposed-bias term improves accuracy, demonstrating that the method's benefits extend beyond ternary quantization.
>
> | Methods     | ARC_e  | ARC_c  | HelS | PIQA   | WinG | AVG     |
> |-----------|--------|--------|-----------|--------|------------|---------|
> | SEQ (2 Bit)      | 0.5909 |0.2449 |   0.3652 |	0.6708	| 0.5312	| 0.4806|
> | Tequila + SEQ (2 Bit)   | 0.6035| 0.2577 |  0.3629| 0.7029| 0.528| 0.491|
>
>
> > **W2 & Q2:** Since the current mechanism focuses only on reactivating dead weights (quantized to 0), it is not evident whether a similar strategy can be applied to weights that are quantized to ±1 in ternary quantization, or other levels in higher-bit setups. Have the authors considered adding similar learnable correction biases globally or per-group even to non-zero weights in ternary quantization?
>
> **A:** Thank you for this interesting suggestion. Our primary contribution is identifying and solving the specific "deadzone trapping" problem. Extending the bias correction to all weights may change the scope of our work into a general quantization error reduction method. In this paper, we focused on the deadzone trapping as the primary bottleneck for ternary QAT stability and efficiency. **The proposed extension is a valuable and complementary direction for future research.**

---

### Official Review · Reviewer_7ZxG · 2025-10-27

**Soundness:** 3
**Presentation:** 4
**Contribution:** 4
**Rating:** 8
**Confidence:** 5

**Summary:**

The paper builds upon the observation that in extreme quantization settings (i.e., weights taking only values -1,0,1) there is a certain area of values that are mapped to zero. The paper then proposes to re-purpose those weights as biases, through which they receive a more stable gradient signal.

**Strengths:**

Experimental results are very promising. The proposed approach excels in all eval benchmarks for the analyzed 3B model, and in all but one case for the 1B model.

The paper further provides analysis of loss trajectory (constant improvement) resource requirements (no extra resources needed).

The visuals of the paper are excellent; very clear and accessible.

A sensitivity analysis of the reactivation threshold is included.

Ablation study considering all components of the approach shows consistent gains.

**Weaknesses:**

Results limited to 1B and 3B models. Other research in quantization-aware training ("The era of 1-bit LLMs") suggests that – at ascale – performance decrease is marginal anyways.

There are other approaches that tackle the same (or highly related) issue of deadzones, e.g., AbsMedian (arXiv:2407.09527) – which do not harm the soundness of the paper but could be considered.

**Questions:**

Are the experimental results only based on a single run or have they been repeated with different random seeds?

---

> ### Author Response · Authors · 2025-11-25
> **Response by Authors (Part 1)**
>
> We sincerely thank Reviewer 7ZxG for the valuable feedback on our manuscript. We are encouraged that they found **our work has  promising results  and excellent visuals.** We will carefully address each concern and revise the manuscript accordingly
>
> > **W1:** Results limited to 1B and 3B models. Other research in quantization-aware training ("The era of 1-bit LLMs") suggests that – at a scale – performance decrease is marginal anyways.
>
> **A:** We appreciate this perspective. While massive models can exhibit resilience, this often comes at an extraordinary computational cost for QAT. Our focus is on making QAT efficient and effective for practical edge-scale models (<4B). To strengthen generalization, we conducted additional experiments on Qwen-4B. Tequila achieves a 3.4% average accuracy gain over the SOTA Absmean, as shown below:
> | Model     | Arc_E  | Arc_C  | Hellaswag | PIQA   | Winogrande | AVG     |
> |-----------|--------|--------|-----------|--------|------------|---------|
> | BF16      | 0.8186 | 0.5213 |    0.5423 | 0.7769 |     0.6772 | 0.66726 |
> | Absmean   | 0.6915 | 0.3507 |    0.4616 | 0.7339 |     0.5856 | 0.56466 |
> | Tequila   | 0.7538 | 0.4189 |    0.4681 | 0.7383 |     0.6133 | 0.59848 |
>
> Critically, a significant performance gap remains at the 4B scale versus FP16. This confirms that the quantization challenge is very much present at edge scales, and Tequila provides an effective solution by mitigating deadzone trapping to bridge this gap efficiently.
>
> >**W2:** There are other approaches that tackle the same (or highly related) issue of deadzones, e.g., AbsMedian (arXiv:2407.09527) – which do not harm the soundness of the paper but could be considered.
>
> **A:** Thank you for pointing out this relevant work. We have evaluated AbsMedian on our 1B/3B models. The results in following table (also in Table 1) shows that found AbsMedian underperforms AbsMean, aligning with the cited paper's observation that it is harder to optimize. Crucially, when combined with our Tequila, AbsMedian's performance is maintained at the most benchmarks. This demonstrates that Tequila's weight reactivation effectively mitigates optimization challenges and is complementary to various base quantization methods.
>
> | Methods             | Size |  Arc_E |  Arc_C | Hellaswag |  PIQA | Winogrande |  GPQA |  AVG  |
> |---------------------|:----:|:------:|:------:|:---------:|:-----:|:----------:|:-----:|:-----:|
> | Absmean             |  1B  |  0.603 | 0.259  |   0.360   | 0.683 |    0.541   | 0.227 | 0.445 |
> | AbsMedian           |  1B  |  0.567 |  0.251 |   0.339   | 0.674 |    0.533   | 0.222 | 0.431 |
> | Tequila + AbsMean   |  1B  | 0.645  | 0.305  |   0.391   | 0.710 |    0.542   | 0.232 | 0.471 |
> | Tequila + AbsMedian |  1B  |  **0.645** |  **0.308** |   **0.393**   |**0.719** |    **0.558**   | **0.237** |**0.477** |
> | Absmean             |  3B  |  0.672 |  0.329 |   0.439   | 0.735 |    0.582   | 0.301 | 0.510 |
> | AbsMedian           |  3B  |  0.636 |  0.299 |   0.406   | 0.713 |    0.558   | 0.181 | 0.466 |
> | Tequila + AbsMean   |  3B  |  **0.702** |  **0.346** |   **0.464**   | **0.739** |    **0.627**   | **0.303** | **0.530** |
> | Tequila + AbsMedian |  3B  |  0.701 |  0.345 |   0.449   | 0.733 |    0.611   | 0.237 | 0.513 |
>
> >**Q1**: Are the experimental results only based on a single run or have they been repeated with different random seeds?
>
> **A:** All results are based on three independent runs with different random seeds. We report the average accuracy, and the experimental section has been updated to reflect this. For clarity, we have also added standard deviation as shaded areas and error bars to the figures.

---

### Official Review · Reviewer_ckpi · 2025-10-29

**Soundness:** 1
**Presentation:** 2
**Contribution:** 3
**Rating:** 2
**Confidence:** 3

**Summary:**

This paper introduces Tequila, a precision enhancement strategy specifically designed for ternary Quantization-Aware Training (QAT) of Large Language Models (LLMs). The paper argues that the primary cause of accuracy degradation in ternary quantization is the phenomenon where many weights are quantized to zero, leading to a "deadzone trapping" issue during gradient updates. This trapping prevents these weights from being effectively updated. To address this, the authors propose reutilizing these weights trapped in the deadzone by treating them as dynamic biases, thereby making their gradient updates more effective and meaningful. Through this approach, the authors claim their method can surpass current SOTA techniques for ternary quantization. Furthermore, the authors provide ablation studies in a progressive manner to illustrate the effectiveness of their proposed techniques.

**Strengths:**

1.  **Valuable Research Problem:** Improving the accuracy of ternary quantization is highly beneficial for model acceleration during inference, as it allows replacing complex multiplications with additions. This aligns well with current trends in model quantization research. Furthermore, the paper's experimental results demonstrate a significant accuracy improvement compared to other existing ternary quantization works of similar scale.
2.  **Sufficient Experiments and Impressive Results:** Although the authors conducted experiments on relatively small model scales (1B/3B) and a token scale of 10B, they compared their method extensively against numerous other ternary quantization works of the same scale and achieved impressive accuracy gains. Additionally, they performed comprehensive ablation studies concerning method design, parameter selection, and quantization granularity, yielding complete results. Their experimental outcomes are exceptionally good; other ternary-quantized models of similar scale, even after training on 100B tokens, often fail to match the performance on most tasks achieved by TequilaLLM trained on only 10B tokens.

**Weaknesses:**

1.  **Theoretical Analysis is Unconvincing and Contains Flaws:**
    1) In Section 2, the paper primarily states the drawbacks of traditional gradient computation for ternary quantization problems. However, the gradient formula (3) presented in line 132 is difficult to further derive into the authors' statement about the gradient computation flaw (line 152: *"Because they contribute no information to the prediction loss L, they receive uninformative gradients during backpropagation"*), and the authors do not provide further theoretical analysis supporting this statement. Formula 3 for gradient calculation seems to originate from the reference (Chen et al., 2024). However, if using a traditional STE estimation scheme, the calculation of the original weight gradient ∂L/∂wᵢ should be identical to the quantized weight gradient ∂L/∂Ŵ, which is (∂L/∂Y) * xᵢ * α. In this case, when the original weight lies within the range (–δ, δ) (the "deadzone" scenario claimed in the paper), the weight gradient would be the same as outside the deadzone. Therefore, the paper's derivation of the "DEADZONE TRAPPING" scenario based solely on this formula in Section 2.2 is unconvincing. Furthermore, line 161 uses exaggerated terms like "permanently inactive" and "drastically" to describe the harm of the deadzone, which is unpersuasive.
    2) Section 3's introduction of the bias-like correction term is even more confusing. Firstly, regarding the first variant, Minima Reactivation, the so-called "informative" value mentioned in line 192 is merely sign(x)*ε from a formulaic perspective, and the authors do not provide complete analysis proving this design is effective. In my view, this approach seems more like splitting zero into a small positive and a small negative value to expand capacity, effectively simulating 4-bit quantization, which naturally performs better than 3-bit.
    3) Section 3.2's proposal of the Tequila method is essentially a relaxation of STE approximation, a concept already present in quantization literature. The step-by-step simplification process from STE -> differentiable reactivation -> activation-free bias (using dead weights as bias) -> letting these weights act as both bias and participate in multiplication (dual role) seems reasonable at each step, but lacks persuasive power and involves strong logical leaps. For example, in line 246, the authors claim "bias-like term can be precomputed offline, reducing inference overhead to nearly zero," but do not explain: Why can this term be completely decoupled from the input? Under different input distributions, is it truly "approximately constant"? This is an empirical assumption without mathematical or experimental support. Ultimately, in the derived gradient correction formula, the authors claim in line 165 that this gradient for dead weights is "superior," but provide no quantitative analysis proving that adding a constant bias term xᵢ in formula (8) results in a better gradient estimation than normal STE.
2.  **Exaggeration is Prevalent in the Text:** For example, line 152 asserts that weights within (–∆, ∆) under STE have gradients that "contribute **no information** to the precision loss L" and line 153 uses **'uninformative'**, line 220 directly claims "**fundamentally** enable the efficient recovery of previously trapped weights, **dramatically** accelerating convergence", and the advantages given in line 270 seem to be summarizing for the sake of summarizing, with little relation to the actual logical flow of the text. The paper uses a large number of highly positive adjectives to describe their method (fundamentally, informative, superior), but lacks specific data or theoretical support, giving the impression of "more promotion than argumentation."
3.  **Insufficient Analysis of Deadzone Generality:** According to the paper's logic, the deadzone phenomenon should exist for quantization at any bit-width, as some weights will be quantized to zero. If the authors could provide a more detailed analysis of this deadzone problem across different bit-width quantization scenarios, it would be better.
4.  **Lack of Specific Details on the Mixed-Precision Framework Design:** It appears that during the quantization training process, apart from the online quantization of weights to 3-bit, other components such as main weights, activations, gradients, and optimizer states should remain at higher precision. The authors should briefly state this in the main text or at least in the appendix, explaining that maintaining these parts at high precision is also necessary to ensure the final ternary-quantized model achieves good results during inference, to avoid misleading readers unfamiliar with this field.

**Questions:**

The main questions focus on the paper's theoretical analysis, as detailed in weakness 1. I'll briefly summarize here to facilitate the author's rebuttal:

1.  Can the authors provide a more reasonable and complete analysis explaining *why* gradient updates within the deadzone are noisy, uninformative, and potentially lead to "making model's parameters **permanently inactive**" (line 161)?
2.  Can the authors provide a more logical reasoning process for Section 3.1, primarily addressing the origin and justification of the bias-like design term?
3.  Can the authors add more rigorous, step-by-step deductions in Section 3.2 when deriving the final Tequila formula to ensure soundness?

Additionally, there are a few other minor questions:

4.  In Table 1, were the experiments conducted using exactly the same datasets across all compared methods?
5.  The results in Table 2 are particularly impressive. It's striking that LLM-QAT, BitNet, and Spectra, trained on 100B tokens, perform worse on most tasks than TequilaLLM trained on only 10B tokens. Have the authors considered and can they discuss the fundamental reasons behind this impressive phenomenon?

---

> ### Author Response · Authors · 2025-11-25
> **Response by Authors (Part 1)**
>
> We appreciate Reviewer ckpi's detailed and insightful feedback, especially recognizing that **our paper conducts sufficient experiments and gets impressive results.** We have addressed all raised concerns in the points below and have incorporated these revisions into the updated manuscript.
>
> >**Q1 & W1.1**: In traditional STE, when the original weight lies within the range $(-\Delta, \Delta)$, the weight gradient would be the same as outside the deadzone. Can the authors provide a more reasonable and complete analysis explaining why gradient updates within the deadzone are noisy, uninformative, and potentially lead to making model's parameters permanently inactive?
>
> **A:** While traditional STE assigns the same algebraic gradient ($\frac{\partial L}{\partial Y} x_i$) for active and dead weights, **their optimization dynamics differ due to distinct roles in the forward pass.** The critical issue is that **dead weights ($\hat{w}_i = 0$) and their corresponding inputs $x_i$ contribute less to the output $Y$ and loss $L$.** Consequently, the upstream signal $\frac{\partial L}{\partial Y}$ reflects less sensitivity to $w_i$ and $x_i$, making the overall gradients $\frac{\partial L}{\partial Y} x_i$ noisy, less informative signals (we change the term “uninformative” to “less informative”).
>
> A Simplified Example: Consider $Y = x_1w_{\pm 1} + x_2 w_0 \approx  \alpha x_1 \hat w_{\pm 1} + x_2 \hat w_0   $, where $w_{\pm 1} $ is quantized to $\hat w_{\pm 1} = \pm 1$ and $w_{0}$  is a dead weight ($\hat w_{0} = 0$). So the $\hat Y = \alpha x_1\hat w_{\pm 1}$, which is the real output. The gradient of $w_{0}$ is $\frac{\partial L}{\partial w_0} = \frac{\partial L}{\partial \hat Y} x_2$. Since the $w_0$ and $x_2$  provide no information to the $L$ and $\hat Y$, the gradients $ \frac{\partial L}{\partial \hat Y}$ is less correlated to them. Under the common assumption that $E[x] \approx 0$, the $ E[\frac{\partial L}{\partial w_0} ]= E[\frac{\partial L}{\partial \hat Y} x_2] \approx E[\frac{\partial L}{\partial \hat Y}]E[x_2] \approx 0$ resulting in a gradient with an approximately zero mean. Combined with high variance from the lack of correlation, this makes the gradients for dead weights inherently noisy and less informative.
>
> These noisy gradients prevent dead weights from accumulating consistent update signals, trapping them within the deadzone.  When a large number of weights remain at deadzone for an extended period, the model easily becomes trapped in a suboptimal state. When some weights occasionally break free, their abrupt changes often trigger opposing gradients that pull them back. This dynamic results in weights accumulating at the deadzone boundary, **trapped in a cycle of ineffective oscillation,** as shown in Fig.1 (top) and Fig. 10. Ultimately, deadzone trapping renders a large number of the model weights long-term inactive, severely impairing both model capacity and training efficiency. (we change the term “permanently” to “long-term”).
>
> We empirically validate this effect by measuring the average Gradient Signal-to-Noise Ratio (GSNR) [1] for dead and active weights during training (using the Standard STE). As shown in Fig. 3 (left) and the following summary Table, dead weights exhibit lower GSNR than active ones in the early training stage. This confirms that their gradients are noisier and less informative. Furthermore, after an extended period, the GSNR for both weight types drops to a consistently low level. This indicates that the model has entered a state of deadzone trapping, where the persistent oscillation of weights at the boundary severely impedes further optimization.
> | Training steps | Active Weight (AbsMean)  | Dead Weight (AbsMean) |  Active Weight (Minima Reactivation)  | Reactive Weight (Minima Reactivation) |Active Weight (Tequila) | Reactive Weight (Tequila) |
> |---|--|--|--|--|--|--|
> | 0|     0.169  |0.156|    0.739 |0.873|0.432 |0.431|
> | 200|    0.458  |0.384|    0.769 |0.80|0.943 |0.935|
> | 400|     0.257  |0.223|     0.570  |0.579|0.974 |0.965|
> | 600|     0.087  |0.081|     0.635  |0.622|0.958 | 0.952|
> | 800|     0.099  |0.100|     0.710  |0.680|0.949 | 0.942|
> | 1000|     0.097  |0.108|     0.728  |0.690|0.802 |0.795|
>
> Table: GSNR for the weights during the training
>
> [1] Liu, Jinlong, et al. "Understanding why neural networks generalize well through gsnr of parameters." arXiv preprint arXiv:2001.07384 (2020).

---

> > ### Author Response · Authors · 2025-11-25
> > **Response by Authors (Part 2)**
> >
> > >**Q2 & W1.2**: In my view, this approach (Minima Reactivation) seems more like splitting zero into a small positive and a small negative value to expand capacity, effectively simulating 4-bit quantization, which naturally performs better than 3-bit. Can the authors provide a more logical reasoning process for Section 3.1, primarily addressing the origin and justification of the bias-like design term?
> >
> > **A:** Minima Reactivation is NOT intended to increase representational precision (that's why we deliberately use minima constant mapping instead of multiplication, making it behave differently from a higher-bit scheme). **Its mainly effect comes from providing a more meaningful forward signal and a more meaningful gradient path for dead weights.**
> >
> > As discussed above, deadzone trapping issue comes from less contribution of dead weights to the output. Our motivation is to allow dead weights to contribute the forward output, even with minimal but informative values, which can break trapping by receiving a more direct and informative gradient signal. Therefore, Minima Reactivation scheme, the bias-like term in Eq. (5) is the natural consequence, which let the dead weight provide sign signals to direct contribution to the output $Y$. As a result, gradients become more related to the weight $w_i$ and less noisy.
> >
> > The  GSNR measurements in Fig. 3 (middle)  and above table confirm this: reactivated weights exhibit smaller GSNR gaps to active weights compared to dead weights, showing that Minima Reactivation provides more informative gradients.
> >
> > > **Q3 & W1.3:** Can the authors add more rigorous, step-by-step deductions in Section 3.2 when deriving the final Tequila formula to ensure soundness? For example. Why can this term be completely decoupled from the input? Under different input distributions, is it truly "approximately constant”?
> >
> > **A:** We have thoroughly reorganized Section 3.2 to provide a more rigorous, step-by-step deduction of the Tequila formula, clarifying the rationale behind each simplification. The refined derivation process is outlined as follows:
> >
> > **Differentiable Reactivation:** The core purpose of this step is to remove the STE and provide direct gradients to dead weights. A fixed minimal mapping is non-differentiable with respect to the weight. Therefore, we parameterize $\epsilon$  as a differentiable function of the original weight $\lambda w_i$. This critical design ensures that the gradient $\partial Y / \partial w_i $ can flow directly through $\lambda w_i$. providing a clean, non-STE path to update the dead weight itself based on its contribution to the loss.
> >
> > **Activation-Free Bias and Simplification:** We update the previous claim as following: the primary purpose of introducing an activation-free bias is to convert the online computation into an offline one. This simplification is justified by two reasons. First, it remains faithful to our core motivation of enabling dead weights to contribute to the output; a bias term achieves this in an activation-free manner, perfectly aligning with our hypothesis while minimizing overhead. Second, we find empirical support for this approximation: the cosine similarity between the original bias-like and the simplified bias is high (>70%), as shown in Fig. 4, validating that our simplification retains the essential functional behavior.
> >
> > **Hybrid Roles of Reactivated Weights:** While the activation-free bias efficiently reactivates dead weights, it completely discards input information. To address this, we assign reactivated weights to hybrid roles. These weights are simultaneously maintained as participants in bias and the ternary matrix multiplication. This dual role creates a mixed gradient from both the standard ternary pathway and the direct bias pathway, thereby preserving input-dependent information and delivering a direct, informative gradient signal to enable effective optimization. As shown by the GSNR measurements in Fig.3 (right) and above Table, Tequila provides high and stable GSNR during the training, thereby enabling more efficient optimization.
> >
> > > **W2:** Many strong claims (“permanently inactive”, “drastically”, “fundamentally enable…”) appear unsupported by rigorous proof or quantitative evidence
> >
> > **A:** We thank the reviewer for highlighting the need for precise language. In response, we have carefully revised the manuscript to temper strong claims and ensure all statements are backed by evidence.
> >
> > Specifically, we have replaced or softened overstated adjectives throughout the text. For instance, "drastically", "fundamentally", "significantly" have been removed, "permanently inactive" has been changed to "long-term inactive," and "uninformative" has been revised to "less informative." These adjustments are directly supported by GSNR analysis of gradient dynamics in Figure 3.

---

> > > ### Author Response · Authors · 2025-11-25
> > > **Response by Authors (Part 3)**
> > >
> > > > **W3:** If the authors could provide a more detailed analysis of this deadzone problem across different bit-width quantization scenarios, it would be better.
> > >
> > > **A:** In higher-bit (e.g., 2-bit, 4-bit) regimes, the paradigm is multiplication-based. These schemes typically do not require a zero represenzation and can effectively mitigate the deadzone-trapping problem through zero-point shifting [2]. In contrast, ternary quantization uniquely fixes the zero point to remove multiplications, which creates the deadzone trapping analyzed in our paper. This paper’s primary contribution lies in identifying the deadzone trapping problem specific to ternary models
> > >
> > > Though other bit schemes do not suffer from deadzone trapping problem, the idea of repurposing weights as bias can still enhance model performance . We applied this idea to 2-bit SEQ by adding a repurposed-bias term $Y = XW +  \sum_i \lambda W_i$. As shown in the table below and in Table 5 of the appendix, Tequila also improves accuracy in the 2-bit case, demonstrating that the mechanism generalizes beyond ternary quantization even when zero-point shifting is already available.
> > >
> > > | Methods     | Arc_E  | Arc_C  | Hellaswag | PIQA   | Winogrande | AVG     |
> > > |-----------|--------|--------|-----------|--------|------------|---------|
> > > | SEQ (2 Bit)      | 0.5909 |0.2449 |   0.3652 |	0.6708	| 0.5312	| 0.4806|
> > > | Tequila + SEQ (2 Bit)   | 0.6035| 0.2577 |  0.3629| 0.7029| 0.528| 0.491|
> > >
> > > > **W4:** The authors should briefly state this (quantization training process) in the main text or at least in the appendix to avoid misleading readers unfamiliar with this field.
> > >
> > > **A:** We have added a brief introduction to the standard (QAT) process in the background section of the main text, and a detailed description of the training procedure has been included in the Appendix C.
> > >
> > > > **Q4:** In Table 1, were the experiments conducted using exactly the same datasets across all compared methods?
> > >
> > > **A:** Yes, all experiments in Table 1 were conducted using **same training datasets and hyperparameter settings**.
> > >
> > > >**Q5:** The results in Table 2 are particularly impressive. It's striking that LLM-QAT, BitNet, and Spectra, trained on 100B tokens, perform worse on most tasks than TequilaLLM trained on only 10B tokens. Have the authors considered and can they discuss the fundamental reasons behind this impressive phenomenon?
> > >
> > > **A:** We appreciate the reviewer's insightful observation. The key reason TequilaLLM outperforms methods trained on 100B tokens is optimization quality. Even with 100B tokens, existing methods all use traditional tenary quantization methods,  suffering from deadzone trapping issue, is easy to fall into suboptimal local minima because their dead weights remain long-term inactive, trapped in a cycle of ineffective oscillation around the deadzone boundary, as shown in Figure 3 (left) and Figure 10.
> > >
> > > TequilaLLM eliminates this bottleneck by reactivating dead weights and restoring meaningful gradient flow, allowing these weights to move smoothly back into the active region. This prevents the model from getting stuck in poor minima and allows it to use its capacity more effectively. As a result, TequilaLLM converges to a better optimum with only 10B tokens.
> > >
> > > [2] Liu, Zechun, et al. "ParetoQ: Improving Scaling Laws in Extremely Low-bit LLM Quantization." The Thirty-ninth Annual Conference on Neural Information Processing Systems.

---

> > > > ### Comment · Reviewer_ckpi · 2025-11-26
> > > >
> > > > Thank you very much for the authors’ detailed responses. Most of my questions have been satisfactorily addressed.
> > > >
> > > > Regarding the theoretical analysis, I appreciate the substantial revisions made to the methodological section, which have considerably improved the soundness and persuasiveness of the work. The proposal in Section 2 to use GSNR as a measure of gradient stability is particularly insightful, and the empirical evidence provided lends strong support to the paper’s claims. **I have one minor question:** Appendix E.2 notes that the GSNR values in Figure 3 are estimated using 1,024 sampled training examples. Would it be feasible to monitor GSNR stability throughout training using all samples, rather than relying on a fixed subset?
> > > >
> > > > For the stepwise introduction of the Tequila approach in Section 3, I am basically satisfied with the revised exposition. The clarified logic and the additional Figure 3 indeed strengthen the justification for the role of Reactivated Weights.
> > > >
> > > > I would also like to highlight that the authors have taken my earlier suggestions seriously by removing or softening several strong claims (e.g., “permanently inactive,” “drastically,” “fundamentally enable…”), and ensuring that all assertions are now backed by evidence rather than stated without support. **This practice is essential for research papers—every claim should be grounded in concrete evidence, which is indispensable for the overall soundness of the work.**
> > > >
> > > > Regarding whether the deadzone issue extends to higher-bit quantization, I would like to restate that my question primarily concerns higher-bit formats such as INT4/INT8/FP4/FP8 (often used in PTQ or non–weight-only quantization), which typically include the value 0—unlike binary quantization {−1, +1} or quaternary quantization {−2, −1, +1, +2}. According to the claims in this paper, these quantized values might also fall into relatively small “deadzones,” though this phenomenon would differ significantly from the extreme low-bit QAT settings explored in this paper and in [2] (ParetoQ). Thus, **I am curious whether the “deadzone-trapping-free” insight proposed here could also benefit quantization accuracy in higher-bit settings.**
> > > >
> > > > Overall, I believe the revisions made during the rebuttal phase will help readers (including myself) better understand the Tequila method. I have updated my rating accordingly. Good luck!

---

> > > > > ### Author Response · Authors · 2025-11-28
> > > > >
> > > > > Thank you to Reviewer ckpi for raising our score to 6. We appreciate your valuable comments, which have helped make our paper more solid. Below, we have provided responses to your follow-up questions before rolling back
> > > > >
> > > > > > **Q1** Appendix E.2 notes that the GSNR values in Figure 3 are estimated using 1,024 sampled training examples. Would it be feasible to monitor GSNR stability throughout training using all samples, rather than relying on a fixed subset?
> > > > >
> > > > > **A** It is not feasible to monitor GSNR stability throughout training using all samples, as it require to compute gradients from all data for every steps
> > > > >
> > > > > > **Q2**  whether the “deadzone-trapping-free” insight proposed here could also benefit quantization accuracy in higher-bit settings.
> > > > >
> > > > > **A** We are conducting experiments on INT4 quantization, while it needs take sometime. We will report the results before deadline and after the roll back of the rebuttal.
> > > > >
> > > > > Once again, we thank Reviewer ckpi for the valuable feedback and raising the score to 6

---

> ### Author Response · Authors · 2025-11-29
> **Official Comment by Authors**
>
> Although Reviewer ckpi is unable to reply to our response, we are happy to fulfill our promise and address the follow-up concerns:
>
>  > **Q2** Higher-bit formats such as INT4/INT8/FP4/FP8 might also fall into relatively small “deadzones,”. Whether the “deadzone-trapping-free” insight proposed here could also benefit quantization accuracy in higher-bit settings.
>
> **A**:    We further evaluate Tequila's generalization on  high-bit quantization scheme by applying it to INT4 LSQ quantization. As shown in Figure 11 _(left)_, the deadzone trapping effect in INT4 LSQ is less severe than in ternary quantization (Figure 1). Despite this, Tequila effectively mitigates the remaining trapping, leading to a more stable weight distribution Figure 11 _(right)_. The resulting performance gains, reported in following table (also in Table 5), are consistent but modest. This is because the INT4 LSQ baseline is already highly competitive with the BF16 baseline, leaving little room for large improvement. These results confirm that Tequila generalizes to higher-bit quantization, even when absolute gains are limited by a strong baseline. A comprehensive evaluation of Tequila's performance on higher-bit quantization is provided in Appendix E.3.
> | Methods     | ARC_e  | ARC_c  | HelS | PIQA   | WinG | AVG     |
> |-----------|--------|--------|-----------|--------|------------|---------|
> | BF16   | 0.654 |0.313 |0.477 |0.742 |0.603 |0.557|
> | LSQ (4 Bit)    | 0.670| 0.310| 0.458| 0.741 |0.587| 0.553|
> | Tequila + LSQ (4 Bit)   | **0.672** |**0.313** |**0.459**| **0.747** |**0.593** |**0.557**|
>
> -----
>
> Finally, we would like to sincerely thank Reviewer ckpi once more. Their valuable feedback and rigorous review have significantly improved our paper. Their raising the score to a 6 (althougth it is reverted 2 now) was also a great encouragement to us.

---

### Official Review · Reviewer_GbUj · 2025-11-01

**Soundness:** 3
**Presentation:** 3
**Contribution:** 3
**Rating:** 6
**Confidence:** 3

**Summary:**

This paper presents Tequila, a ternary quantization method for large language models (LLMs) targeting edge deployment. It tackles the “deadzone trapping” problem, where many weights become inactive due to noisy gradients. Tequila reuses these inactive weights as adaptive biases, restoring useful forward and backward signals. Experiments on five benchmarks and two LLaMA sizes show that Tequila achieves higher accuracy and efficiency than existing ternary methods, approaching full-precision performance with much lower hardware and data costs.

**Strengths:**

Clear identification of the core problem: The paper accurately pinpoints “deadzone trapping” as a key obstacle to effective ternary quantization, and Figure 8 clearly demonstrates the effectiveness of the proposed Tequila method in addressing this issue.
Practicality and efficiency: Tequila achieves near–full-precision performance while remaining hardware-friendly, improving accuracy by over 4% compared to SOTA methods and maintaining comparable inference speed to classic ternary quantization approaches such as BitNet.
High-quality and well-presented figures: For example, Figure 1 (“Deadzone Trapping in Ternary Quantization”) and Figure 2 (“Tequila Reactivation Strategy”) intuitively illustrate the problem and solution, while Figure 3 clearly shows faster convergence.
Good reproducibility and transparency: Experimental details and code implementations are publicly available, enabling community verification and reproduction.

**Weaknesses:**

Limited discussion of related work: As a QAT approach, the paper overlooks several concurrent efficient ternary quantization methods such as OneBit[1], as well as recent PTQ ternary approaches like PTQTP[2].
Limited model scale and architectural generalization: Experiments focus mainly on smaller LLaMA models (1B and 3B). Validation on larger models (e.g., 7B, 13B) or different architectures is missing, leaving generalization untested.
Insufficient analysis of λ sensitivity: Although Figure 6 is provided, the paper does not explore how λ interacts with other hyperparameters or training configurations. Like, whether model size affects the optimal λ during training.
[1]Xu Y, Han X, Yang Z, et al. Onebit: Towards extremely low-bit large language models[J]. Advances in Neural Information Processing Systems, 2024, 37: 66357-66382.
[2]Xiao H, Yang R, Yang Q, et al. PTQTP: Post-Training Quantization to Trit-Planes for Large Language Models[J]. arXiv preprint arXiv:2509.16989, 2025.

**Questions:**

Q1: In Section 3.2, the paper states, “Given the approximately symmetric distribution of inputs (activations) in transformer architectures.” Is there any prior literature or empirical evidence supporting this assumption?

Q2: Also in Section 3.2, the paper mentions, “In addition to functioning as an adaptive bias.” However, shouldn’t $\sum_{i\in D}\lambda w_i$ be input-independent? Could the authors clarify this reasoning?

Q3: The paper claims “a learnable reactivation parameter λᵢ for each deadzone weight wᵢ,” but in experiments a fixed λ value is used. Could the authors elaborate on this inconsistency?

Q4: It would strengthen the paper if the authors could include additional comparisons between Tequila and PTQ-based ternary quantization algorithms in terms of quantization efficiency and downstream task performance?

---

> ### Author Response · Authors · 2025-11-25
> **Response by Authors (Part 1)**
>
> First, we would like to thank Reviewer GbUj for finding **our paper is clear, practical with high-quality and well-presented figures.** In the following, we will carefully address each concern raised by Reviewer GbUj and report the changes done to the manuscript:
>
> > **W1 & Q4:** Limited discussion of related work: As a QAT approach, the paper overlooks several concurrent efficient ternary quantization methods such as OneBit[1], as well as recent PTQ ternary approaches like PTQTP[2]. It would strengthen the paper if the authors could include additional comparisons between Tequila and PTQ-based ternary quantization algorithms in terms of quantization efficiency and downstream task performance?
>
> **A:** We thank the reviewer for pointing this out. OneBit is a binary quantization method rather than a ternary QAT approach, but we agree that PTQ-based ternary quantization methods such as PTQTP should be discussed. We have included PTQTP in the related work section B.2.
> Since PTQTP is not open-source, we instead evaluate ternary AWQ, a strong publicly available PTQ baseline. The results are shown below (Also in Table 4). Ternary AWQ suffer from highly accuracy degradation on Qwen 4B (>30%), which explain that the PTQ is not main stream for the ternary quantizaiton.
>
> | Methods| ARC_e  | ARC_c  | Hellaswag | PIQA   | Winogrande | AVG     |
> |--|--|--|--|--|--|--|
> | BF16| 0.8186 | 0.5213 |    0.5423 | 0.7769 |     0.6772 | 0.66726 |
> | PTQ (AWQ) | 0.2702 | 0.2244 |    0.2587 | 0.5305 |     0.5122 |  0.3592 |
> | AbsMean   | 0.6915 | 0.3507 |    0.4616 | 0.7339 |     0.5856 | 0.56466 |
> | Tequila   | **0.7538** | **0.4189** |    **0.4681** | **0.7383** |     **0.6133** | **0.59848** |
> Table. Teneray quantization on Qwen3-4B
>
> > **W2:** Limited model scale and architectural generalization: Experiments focus mainly on smaller LLaMA models (1B and 3B). Validation on larger models or different architectures is missing, leaving generalization untested.
>
> **A:** Thank you for this important point. Our focus is on edge-device LLMs, where 1B-3B models are the most relevant. To address generalization, we have conducted additional experiments on the Qwen3-4B model. Tequila continues to outperform the SOTA Absmean, achieving an average accuracy gain of 3.4%. This result, included in above table and Table 4, confirms that effectiveness of Tequila in mitigating deadzone trapping generalizes beyond the LLaMA architecture.
>
> > **W3:** Insufficient analysis of $\lambda$ sensitivity: Although Figure 8 is provided, the paper does not explore how $\lambda$ interacts with other hyperparameters or training configurations. Like, whether model size affects the optimal  $\lambda$ during training.
>
> **A:** We have expanded our  $\lambda$ sensitivity analysis to include the LLaMA-3B model in Fig 8. The new results in following Table confirm that the optimal  $\lambda$ is correlated with model size, with larger models preferring a smaller $\lambda$. This finding is now included in the paper. Importantly, Tequila's performance remains robust across a wide range of  $\lambda$ values for both 1B and 3B models.
>
> >**Q1:** In Section 3.2, the paper states, “Given the approximately symmetric distribution of inputs (activations) in transformer architectures.” Is there any prior literature or empirical evidence supporting this assumption?
>
> **A** Our statement was based on the common observation that LayerNorm/GroupNorm layers in transformers tend to produce activations with approximately zero mean. However, to strengthen our statement about input-agnostic bias, we have revised the  Section 3.2 in the manuscript. We now justify the input-agnostic bias into two reason. First, this design remains faithful to our core motivation of enabling dead weights to contribute to the output while minimizing overhead. Second, we find empirical support for this approximation: the cosine similarity between the original bias-like and the input-agnostic bias is high (>70\%), as shown in Fig. 4, validating that our simplification retains the essential functional behavior.

---

> > ### Author Response · Authors · 2025-11-25
> > **Response by Authors (Part 2)**
> >
> > > **Q2:** Also in Section 3.2, the paper mentions, “In addition to functioning as an adaptive bias.” However, shouldn’t $\sum_{i\in D} \lambda w_i$ be input-independent? Could the authors clarify this reasoning?
> >
> > **A:** The term "adaptive" was intended to highlight a contrast with the fixed constant used in Minima Reactivation. In our case, the bias $\sum_{i\in D} \lambda w_i$ is updated dynamically according to the changing weight magnitudes during training. We acknowledge the potential for confusion and have replaced "adaptive" with "dynamic" in the manuscript for clarity.
> >
> > > **Q3:** The paper claims “a learnable reactivation parameter $\lambda_i$ for each deadzone weight $w_i$,” but in experiments a fixed $\lambda$ value is used. Could the authors elaborate on this inconsisten
> >
> > **A:** We apologize for this oversight; the mention of a "learnable" parameter was an error from an earlier draft. All experiments use a fixed, global $\lambda$. We found that learnable $\lambda_i$ increased optimization complexity and led to poor performance, consistent with the trend that learnable scaling factors (e.g., in LSQ, SEQ) often underperform simpler methods like Absmean in ternary LLMs. The manuscript has been corrected to remove all references to a learnable $\lambda$.

---

### Author Response · Authors · 2025-11-25
**Summary and General Response to the Reviewers**

We sincerely thank all reviewers for their time and valuable feedback. We are greatly encouraged that **all reviewers recognized the impressive performance of Tequila and its benefits for inference acceleration.** In this rebuttal, we have systematically addressed all questions and concerns, including those regarding our GSNR analysis, generalization to higher bit-widths and larger models, and applicability to other architectures. All corresponding revisions and additional experiments have been incorporated into the updated manuscript, highlighted in blue color.

---

### Author Response · Authors · 2025-11-29
**Summary of the Rebuttal by Authors**

We thank all reviewers for reviews, giving all possitive final scores and recognizing Tequila's impressive performance and its benefits for inference acceleration. Below is a summary of our responses to their concerns:

-----
-----
**Reviewer GbUj (initial rating 6, no further response)**

> **W1 & Q4**: limited related work in PTQ.

**A** Added discussion of PTQ in the related work and included new experiments in Table 4.

> **W2:**  Limited model scale and architectural generalization

**A** Added Qwen-3B experiments in Table 4.

 > **W3:** Insufficient analysis of $\lambda$ on differernt model scale

**A**  added  $\lambda$  analysis for a 3B model in the updated Fig. 8.

 >**Q1**:  Evidence for symmetric input distribution.

**A**: Revised Section 3.2 and added Fig. 4 for validation.

 >**Q2:** Clarification on "adaptive" bias.

 **A**: Changed "adaptive" to "dynamic" for clarity (it adapts to the weights).

>**Q3:** inconsistency regarding learnable $\lambda$

**A:** Corrected a typo;  $\lambda$ is fixed, and all references to it being learnable have been removed.

----
----

**Reviewer ckpi (initial rating 2, increasing to 6 by Nov. 26 08:57 am EST after rebuttal, 1 day earlier than the issue reported on Nov. 27, 10:09 AM EST)**

> **Q1 & W1.1** why gradient updates within the deadzone are noisy

**A:** Added a detailed explanation in the response and the Section 2.2,  empirically validated it with GSNR in Fig. 3 (left).

> **Q2 & W1.2** the origin and justification of the Minima reactivation

**A:**  Provided a detailed justification in the response and Section 3.1 and supported it empirically with GSNR in Fig. 3 (middle).

> **Q3 & W1.3** add more rigorous, step-by-step deductions of Tequila in section 3.2.

**A:** Thoroughly reorganized Section 3.2 with step-by-step deductions and empirical validation in Fig. 3 (right).

>**W2:** Many strong claims

**A:** Removed all strong claims or supported them with rigorous proofs/quantitative evidence.

> **W3.** A more detailed analysis of this deadzone problem across different bit-width quantization scenarios

**A.** included new experimental results for 2-bit and 4-bit quantization in Table 5.

> **Q4.**  In Table 1, were the experiments conducted using exactly the same datasets

**A.** Yes

> **Q5.** The fundamental reason behind TequilaLLM's impressive results.

**A.** The trapping-free mechanism prevents the model from converging to poor local minima, leading to more stable and effective training.

-----
-----

**Reviewer ckpi's follow-up questions after increasing score to 6 by Nov. 26, 08:57 am EST**

> **Q1.** Would it be feasible to monitor GSNR stability throughout training using all samples, rather than relying on a fixed subset?

**A.** It is infeasible as requiring a full backward pass over the entire dataset at every step

> **Q2.** Whether the “deadzone-trapping-free” insight could also benefit accuracy in higher-bit.

**A.**  Added INT4 results in Table 5 and Figure 11. Gains are modest due to strong baseline performance.

-----
-----

**Reviewer 7ZxG  (initial rating 8, no further response)**

>**W1:** Results limited to 1B and 3B models.

**A.** 1B-3B are the most relevant for edge deployment; inluded the Qwen-3 experiments in Table 4

> **W2.** Suggestion to consider AbsMedian.

**A.** Added AbsMedian results in Table 1.

> **Q1.** Are the experimental results only based on a single run or have they been repeated with different random seeds?

**A.** All results are based on three independent runs and reported the average accuracy.

----
-----

**Reviewer N2uB  (initial rating 6, no further response)**

> **W1 & Q1:**  Extension to other quantization schemes without a deadzone.

**A.** Repurposing all weights as bias to increase model capacity; Added 2-bit quantization without deadzone results in Table 5.

> **W2 & Q2:** extension Tequila on other levels in higher-bit schemes.

**A.** Proposed extension is a valuable and complementary direction for future research. Added 4-bit quantization results in Table 5.

---

### Meta-Review · Area_Chair_gs1Z · 2025-12-21

**Summary:**

The paper introduces Tequila, a method that mitigates the “deadzone trapping” problem in ternary quantization for large language models by reinterpreting weights quantized to zero as adaptive biases. The method is motivated by the need to improve gradient flow during training while preserving the hardware efficiency inherent in extreme quantization schemes. Empirical evaluations across several benchmarks using LLaMA-based 1B and 3B models demonstrate that Tequila can significantly close the performance gap between ternary and full-precision approaches, achieving over 4% accuracy improvements compared to state-of-the-art methods. The visualizations and ablation studies are well presented and intuitive, and the reproducibility of the experiments is enhanced by publicly available code and detailed experimental protocols. Several reviewers appreciated the paper’s clarity, the innovative perspective on reactivating inactive weights, and the overall practical benefits for low-cost inference on edge devices.

**Reviewer Concerns:**

However, the reviews also highlighted some notable concerns that hinder a unanimous endorsement of the work. Critics pointed out that the theoretical justification, including the derivation of gradient corrections and the assumptions about activation distributions, is not entirely persuasive and often lacks rigorous backing. In particular, the treatment and fixed use of the reactivation parameter λ raised questions about its consistency and sensitivity to hyperparameter interactions. At the same time, some reviewers felt the narrative occasionally overreaches its claims without sufficient empirical or mathematical support. Additionally, the experimental focus on relatively small-scale models, combined with limited comparisons to alternative quantization strategies (such as PTQ-based methods or approaches like OneBit and AbsMedian), raises doubts about the generality and broader applicability of Tequila.

**Reviewer Scores:**

The negative reviewer believes the revisions made during the rebuttal phase will help readers better understand the Tequila method and the updated rating.

---

### Decision · Program_Chairs · 2026-01-26

Accept (Poster)